# Soft Task-Aware Routing of Experts for Equivariant Representation Learning

**Jaebyeong Jeon**    **Hyeonseo Jang**    **Jy-yong Sohn**    **Kibok Lee**
Department of Statistics and Data Science, Yonsei University
{jaebyeong98, jhyeonseo715, jysohn1108, kibok}@yonsei.ac.kr

## Abstract

Equivariant representation learning aims to capture variations induced by input transformations in the representation space, whereas invariant representation learning encodes semantic information by disregarding such transformations. Recent studies have shown that jointly learning both types of representations is often beneficial for downstream tasks, typically by employing separate projection heads. However, this design overlooks information shared between invariant and equivariant learning, which leads to redundant feature learning and inefficient use of model capacity. To address this, we introduce **S**oft **T**ask-**A**ware **R**outing (STAR), a routing strategy for projection heads that models them as experts. STAR induces the experts to specialize in capturing either shared or task-specific information, thereby reducing redundant feature learning. We validate this effect by observing lower canonical correlations between invariant and equivariant embeddings. Experimental results show consistent improvements across diverse transfer learning tasks. The code is available at `https://github.com/YonseiML/star`.

## 1    Introduction

Self-supervised learning (SSL) has emerged as a prominent paradigm for learning representations from large-scale unlabeled data [3, 20, 4, 17]. Among various approaches, invariant representation learning methods generate multiple views of the same instance through transformations such as data augmentations, and encourage these views to map to the same representation, thereby preserving semantic content regardless of the transformations applied. However, enforcing strict invariance can discard augmentation-aware information, such as color and spatial location, which may degrade performance

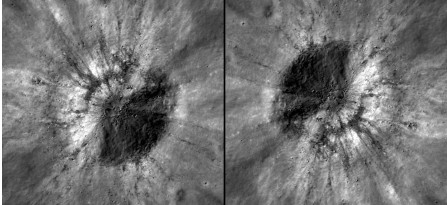

Figure 1: **Crater Illusion.** A lunar image that appear as a dome (left) or a crater (right) depending on orientation [46].

in downstream tasks that depend on such information [27]. To address this limitation, equivariant representation learning has emerged as a complementary paradigm that captures structured variations in the representation space corresponding to transformations applied in the input space [9, 14, 18, 51]. By preserving augmentation-aware information, equivariant representation learning yields richer representations, leading to improved performance on downstream tasks.

Recent approaches such as EquiMod [9] employ a shared encoder followed by two projection heads, which independently produce invariant and equivariant embeddings from a shared representation. This design implicitly assumes that the invariant and equivariant learning tasks are independent, encouraging each projection head to specialize in its respective objective. However, this assumption might not be true in practice; rather, these two tasks are inherently interdependent. This interdependence can be intuitively illustrated by the well-known *crater illusion*, where the same lunar surface may appear either as a crater or a dome depending on the light conditions, as shown in Figure 1. For instance, recognizing whether the object is a crater or a dome—a semantic category typically

39th Conference on Neural Information Processing Systems (NeurIPS 2025).

captured through invariant learning—helps infer which side is illuminated—a property expected to be captured through equivariant learning. Conversely, understanding the orientation of the scene helps recognize the semantic category. This bidirectional dependency highlights the interdependent nature of invariant and equivariant learning.[1] Consequently, when two independent projection heads are employed for invariant and equivariant learning, they tend to redundantly capture shared information across branches—a phenomenon we refer to as *redundant feature learning*.

To mitigate redundant feature learning, we introduce **S**oft **T**ask-**A**ware **R**outing (STAR)—a strategy that explicitly coordinates shared and task-specific information. We instantiate STAR in two forms: (i) by adding a shared projection head that provides a common embedding bridging the two learning objectives and capturing information beneficial to both, and (ii) by adapting the **M**ulti-gate **M**ixture-**o**f-**E**xperts (MMoE) [31] as the projection module, which dynamically allocates experts based on the input for each task. These designs allow the model to more effectively disentangle shared and task-specific information. For the latter implementation, we employ MMoE solely during self-supervised pretraining and transfer only the encoder for downstream tasks, unlike its conventional use for joint training and inference in supervised multi-task learning [31].

Our contributions are summarized as follows:

- We reveal that invariant and equivariant objectives are inherently interdependent, showing that conventional two-branch approaches with separate projection heads independently encode shared information across tasks, leading to redundant feature learning.

- We propose **S**oft **T**ask-**A**ware **R**outing (STAR) for invariant–equivariant representation learning that explicitly coordinates shared and task-specific information, thereby mitigating redundant feature learning and improving transfer performance across various downstream tasks.

- We validate the effectiveness of STAR by demonstrating a substantial reduction in canonical correlation between projection modules, along with a positive correlation between reduced redundant feature learning and improved generalization performance.

## 2 Related Work

**Equivariant Representation Learning.** Early approaches to SSL enforce invariance to transformations by encouraging augmented views of the same image to produce similar embeddings [3, 20, 17, 4], but such constraints may discard informative features like color or spatial location, which could be important for downstream tasks [27]. To address this limitation, recent research explores equivariant representation learning, aiming to learn representations that transform consistently with input transformations. E-SSL [8] introduces an auxiliary task of predicting the applied augmentation, while AugSelf [27] predicts the difference between applied augmentation parameters. Both approaches implicitly learn equivariant representations by predicting input transformations through embedding space differences without explicitly modeling the transformation functions. CARE [18] shows that rotational symmetry in the representation space, aligned with input augmentations, can induce equivariance without modeling the transformations. In contrast, recent methods such as EquiMod [9], SIE [14], and STL [51] explicitly model transformation effects by predicting augmentation-induced shifts in the embedding space. EquiMod achieves this through a learnable predictor conditioned on the applied augmentation. SIE splits the representation into invariant and equivariant parts, and generates augmentation-conditioned predictor using a hypernetwork. STL learns transformation representations directly from unlabeled image pairs. Both STL and CE-SSL [50] are trained with explicit augmentation parameters, while CE-SSL focuses on preserving structured variability in the representation space rather than explicitly modeling the transformations. Our proposed approach, STAR, advances this line of work by emphasizing the interplay between invariant and equivariant learning. Rather than treating the two objectives as independent, STAR explicitly models the shared information essential to both while mitigating redundant feature learning, thereby enabling the model to more effectively capture features distinctive to each objective.

**Mixture of Experts.** The Mixture of Experts (MoE) framework was first introduced by [23], where multiple experts are trained jointly along with a single gating module. This gate performs soft routing by assigning input-dependent weights to each expert's output. The router adaptively combines these outputs, enhancing both flexibility and performance on downstream tasks. Recently, MoE has been

---

[1] In Figure 3, we empirically verify the existence of shared information between invariant and equivariant learning tasks using a benchmark dataset.

extended to multi-task and other representation learning settings. One such example is MMoE [31], which introduces a multi-gate architecture into the MoE framework for multi-task learning. In this design, experts are shared across tasks, while each task is equipped with its own gating network. This setup enables the model to learn task relationships directly from data. In computer vision, V-MoE [42] introduces a sparse Vision Transformer that achieves comparable accuracy to large dense models with about half the inference cost. Neural Experts [1] further extend MoE to implicit representations by dividing the input space among MLP experts, enabling local, piecewise function learning. Unlike conventional MoE models that require routing during both training and inference—making them difficult to transfer across tasks—STAR confines the MMoE structure to the projection head during pretraining. Since projection heads are discarded after self-supervised pretraining, the expert specialization achieved within them does not need to be retained during transfer. This design preserves the benefits of expert routing while completely eliminating the transferability limitation of conventional MoE, enabling efficient fine-tuning and seamless adaptation to downstream tasks.

## 3    Preliminaries: Invariant and Equivariant Representation Learning

Let $\mathcal{X}$ be the image space and $\mathcal{A}$ be a set of augmentation parameters that induce transformations on $\mathcal{X}$. Given an image $x \in \mathcal{X}$ and an augmentation parameter $a \in \mathcal{A}$, we define the transformation function $T : \mathcal{X} \times \mathcal{A} \to \mathcal{X}$ that produces the augmented view $T(x; a)$. An encoder $f : \mathcal{X} \to \mathcal{Y}$ maps the input image to its latent representation $y \in \mathcal{Y}$.

**Invariant Representation Learning.**    Invariant representation learning aims to learn an encoder $f$ that is invariant to transformations applied to the input. Formally, invariance on the representation space is defined as:
$$\forall a \in \mathcal{A}, \quad f(x) = f(T(x; a)). \tag{1}$$
A relaxed formulation, which requires consistency among transformed views of the same image, is given by:
$$\forall a, a' \in \mathcal{A}, \quad f(T(x; a)) = f(T(x; a')). \tag{2}$$
If the parameter of an identity transformation is included in $\mathcal{A}$, then Eq. (1) implies Eq. (2).

In practice, this invariance is achieved by training the encoder to produce similar representations across augmented views of the same image. This is commonly achieved by minimizing a dissimilarity loss of the form:
$$\mathcal{L}^{\text{inv}} = \mathcal{L}(f(T(x; a)), f(T(x; a'))). \tag{3}$$
where $a, a' \in \mathcal{A}$. In contrastive learning, a common choice for the loss function $\mathcal{L}$ is the InfoNCE loss [47], which aims to minimize the distance between representations of augmented views of the same image, while pushing them away from representations of other images. This leads the encoder to learn representations that are consistent across views, thereby preserving semantic content while discarding information that is specific to the applied transformations.

**Equivariant Representation Learning.**    Equivariant representation learning focuses on learning representations that reflect the transformations applied to the input. Unlike invariance, which seeks to map all augmented views to the same representation, equivariance preserves structured changes that correspond to the transformation in the input space. Formally, equivariance on the representation space is defined as:
$$\forall a \in \mathcal{A}, \quad f(T(x; a)) = \phi_T(f(x), a). \tag{4}$$
where $\phi_T : \mathcal{Y} \times \mathcal{A} \to \mathcal{Y}$ denotes a transformation function in the representation space. The function $\phi_T$ is learned to reflect how the input transformation $T(\cdot; a)$ affects the latent representation. When $\phi_T$ reduces to the identity function for all $a$, Eq. (4) simplifies to the invariance condition in Eq. (1). In this case, invariance can be viewed as a special case of equivariance.

To encourage equivariance during training, the model learns to predict the transformed representation of a transformed input. This leads to a learning objective of the form:
$$\mathcal{L}^{\text{eq}} = \mathcal{L}(\phi_T(f(x), a), f(T(x; a))). \tag{5}$$

where the loss $\mathcal{L}$ minimizes the discrepancy between the predicted and target representations of the transformed input. This formulation is designed to capture features that are sensitive to transformations, such as pose and color. It is often incorporated as an auxiliary loss in addition to

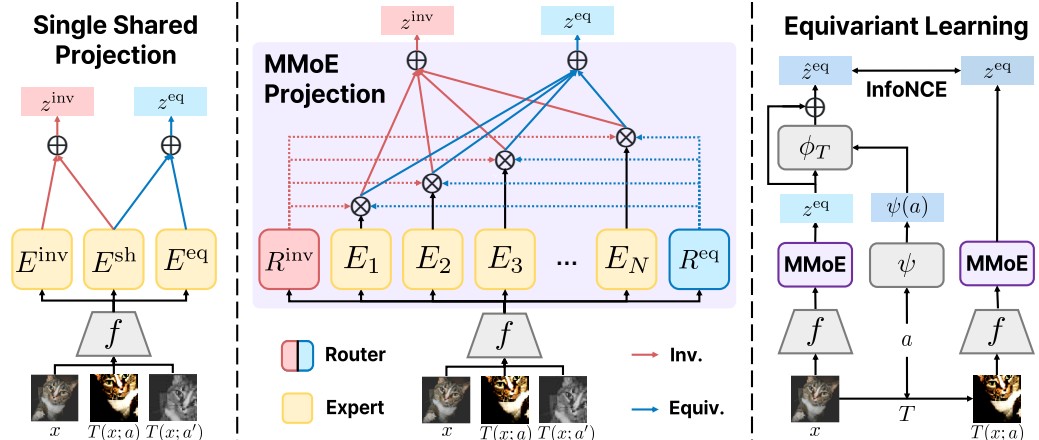

Figure 2: **Overview of Proposed Routing Strategy: Soft Task-Aware Routing (STAR) of Experts.** Given two augmented views $T(x;a)$ and $T(x;a')$, the encoder $f$ extracts features, which are then projected by the single shared projection (with three experts) or the MMoE projection module into invariant ($z^{\text{inv}}$) and equivariant ($z^{\text{eq}}$) embeddings. For equivariant learning, the projected augmentation parameter $\psi(a)$ and the equivariant embedding $z^{\text{eq}}$ are fed into a predictor $\phi_T$ to predict the target embedding $z^{\text{eq}}$. In practice, we implement $\psi$ as a single-layer MLP, and $\phi_T$ as a 3-layer MLP.

the invariant learning objective in recent works on equivariant learning [9, 14, 51]. The overall invariant–equivariant loss is formulated as a weighted sum of the invariant and equivariant losses:

$$\mathcal{L} = \mathcal{L}^{\text{inv}} + \lambda \mathcal{L}^{\text{eq}}. \tag{6}$$

where $\lambda$ is a balancing coefficient that controls the contribution of the equivariant loss.

## 4 Method

In this section, we present Soft Task-Aware Routing (STAR) for learning invariant and equivariant representations. Motivated by redundant feature learning that arises when using separate projection heads for the invariant and equivariant objectives, STAR disentangles shared and task-specific information and adaptively weights their contributions based on the task and input image. For clarity, we refer to each projection head as an expert throughout the paper.

### 4.1 Soft Task-Aware Routing

**Setup.** Given a batch of input images $\{x_i\}_{i=1}^B$, we generate two augmented views per image by applying transformations $T(\cdot; a_i)$ and $T(\cdot; a_i')$, where $a_i, a_i'$ are augmentation parameters sampled from a distribution over $\mathcal{A}$. Specifically, for the $i$-th image $x_i$, we obtain two augmented views:

$$v_i = T(x_i; a_i), \quad v_{i+B} = T(x_i; a_i'). \tag{7}$$

Thus, the resulting batch consists of $2B$ augmented views, where the first $B$ correspond to $\{v_i\}_{i=1}^B$ and the remaining $B$ to $\{v_{i+B}\}_{i=1}^B$. Each augmented view $v_i$ is processed by the encoder $f$ to extract a latent representation. The features are then passed into our proposed task-aware projection modules, as illustrated in Figure 2, to produce the invariant embedding $z_i^{\text{inv}}$ and the equivariant embedding $z_i^{\text{eq}}$.

**Single Shared Projection.** As discussed in Section 1, using two separate experts for invariant and equivariant learning can lead to redundant feature learning on experts. To alleviate this issue, it is crucial to model the shared information between the two tasks. A straightforward yet effective approach is to introduce a single expert shared across invariant and equivariant learning tasks, thereby allowing the model to capture information essential to both. To formalize this, we define three experts taking the representation as input: an invariant expert $E^{\text{inv}}$, an equivariant expert $E^{\text{eq}}$, and a shared expert $E^{\text{s}}$. The embeddings are then computed as follows:

$$z_i^{\text{inv}} = E^{\text{inv}}(f(v_i)) + E^{\text{sh}}(f(v_i)), \quad z_i^{\text{eq}} = E^{\text{eq}}(f(v_i)) + E^{\text{sh}}(f(v_i)). \tag{8}$$

An overview of this computation is illustrated in the left panel of Figure 2. This additive formulation naturally encourages the shared expert to capture information relevant to both tasks, since its output contributes directly to both invariant and equivariant objectives. However, the shared expert's output is always weighted equally regardless of tasks and input images, making the model unable to account for the varying importance of shared information across them. While effective in reducing redundancy, this approach is inflexible and cannot adjust how shared and task-specific information are weighted.

**MMoE Projection.** To extend the single shared projection to a more flexible and adaptive design, we introduce the MMoE projection module, which adaptively selects relevant experts from a shared set based on the task and input. This mechanism is illustrated in the center panel of Figure 2. Specifically, the MMoE module contains the shared set of experts $\{E_k\}_{k=1}^N$ and two task-specific routers $R^{\text{inv}}$ and $R^{\text{eq}}$. The routers compute assignment weights as:

$$s_{i,k}^{\text{inv}} = \text{softmax}_k(R^{\text{inv}}(f(v_i))), \quad s_{i,k}^{\text{eq}} = \text{softmax}_k(R^{\text{eq}}(f(v_i))), \tag{9}$$

where $s_{i,k}^{\text{inv}}$ and $s_{i,k}^{\text{eq}}$ denote the assignment weights of the $k$-th expert for the $i$-th view in the batch, used in the invariant and equivariant projections, respectively. Here, the $k$-th component of the softmax activation applied to a vector $w = [w_1, w_2, ..., w_N] \in \mathbb{R}^N$ is defined as:

$$\text{softmax}_k(w) = \frac{\exp(w_k)}{\sum_{j=1}^N \exp(w_j)}. \tag{10}$$

In our case, $w$ is the router output, and $g_k$ denotes the score assigned to the $k$-th expert. Unlike hard assignment strategies, our method employs soft routing, where each expert contributes proportionally to its routing score. Accordingly, the invariant and equivariant embeddings are computed as weighted sums:

$$z_i^{\text{inv}} = \sum_{k=1}^N s_{i,k}^{\text{inv}} E_k(f(v_i)), \quad z_i^{\text{eq}} = \sum_{k=1}^N s_{i,k}^{\text{eq}} E_k(f(v_i)). \tag{11}$$

The equivariant embedding of the original input $x_i$, denoted as $z_i^{\text{o}}$, is obtained in the same manner:

$$z_i^{\text{o}} = \sum_{k=1}^N s_{i,k}^{\text{eq}} E_k(f(x_i)). \tag{12}$$

This design enables the model to adaptively route experts depending on the learning objective (i.e., invariant or equivariant) and the input image, resulting in a natural division into shared and task-specific roles. Notably, the single shared projection introduced earlier can be regarded as a degenerate case of MMoE projection, where all experts are equally weighted rather than adaptively determined.

### 4.2 Equivariant Learning

To explicitly model the shift induced by augmentations in the equivariant embedding space, we compute the predicted equivariant embedding by adding a shift that reflects the effect of the applied augmentation to the original equivariant embedding $z_i^{\text{o}}$. This shift in the equivariant embedding space is predicted by an equivariant predictor $\phi_T$ that takes as input both $z_i^{\text{o}}$ and a projected augmentation parameter, obtained via a projection function $\psi$:

$$\hat{z}_i^{\text{eq}} = z_i^{\text{o}} + \phi_T(z_i^{\text{o}}, \psi(a_i)). \tag{13}$$

The residual connection in Eq. (13) ensures that the semantic content of the original equivariant embedding is preserved while effectively modeling the shift in the embedding space caused by transformations.

Following the standard formulation of invariant learning, we define the equivariant loss in a similar manner using the InfoNCE loss [47]. For each predicted embedding $\hat{z}_i^{\text{eq}}$, the corresponding target embedding $z_i^{\text{eq}}$ serves as a positive sample, while the remaining equivariant embeddings in the batch serve as negatives. The overall equivariant loss is defined as:

$$\mathcal{L}^{\text{eq}} = -\frac{1}{2B} \sum_{i=1}^{2B} \log \frac{\exp\left(\text{sim}(z_i^{\text{eq}}, \hat{z}_i^{\text{eq}})/\tau\right)}{\sum\limits_{j=1, j \neq i}^{2B} \exp\left(\text{sim}(z_i^{\text{eq}}, z_j^{\text{eq}})/\tau\right)}. \tag{14}$$

where $\text{sim}(\cdot, \cdot)$ denotes cosine similarity and $\tau$ is a temperature parameter. We use the invariant–equivariant loss described in Eq. (6) for our method employing SimCLR [3] for invariant loss unless otherwise specified. We set $\tau = 0.2$ and $\lambda = 1$ for all experiments.

Table 1: **Out-of-Domain Classification.** Linear evaluation accuracy (%) of ResNet-50 pretrained on ImageNet100 for 500 epochs and ResNet-18 pretrained on STL10 for 200 epochs, respectively. **Bold entries** indicate the best performance among methods, while underlined entries denote the second best. Standard deviations are reported in Table B.2.

| Method | CIFAR10 | CIFAR100 | Food | MIT67 | Pets | Flowers | Caltech101 | Cars | Aircraft | DTD | SUN397 | Mean | Avg. Rank |
|---|---|---|---|---|---|---|---|---|---|---|---|---|---|
| | | | | | *ImageNet100-pretrained ResNet-50* | | | | | | | | |
| SimCLR | 87.88 | 67.92 | 63.60 | 66.57 | 76.71 | 88.37 | 85.02 | 47.09 | 48.23 | 69.17 | 52.02 | 68.42 | 5.00 |
| AugSelf | 88.61 | 69.68 | 65.37 | 67.51 | 77.24 | 89.70 | 85.09 | 47.48 | 48.65 | 69.31 | 53.00 | 69.24 | 3.82 |
| EquiMod | 88.99 | 70.22 | 64.43 | 67.54 | 77.78 | 90.33 | 86.62 | 48.94 | 49.91 | 69.33 | 52.79 | 69.72 | 3.18 |
| CARE | 82.81 | 58.97 | 55.78 | 56.39 | 59.89 | 75.84 | 73.14 | 29.12 | 36.00 | 62.22 | 42.48 | 57.51 | 6.00 |
| STAR-SS | 89.81 | 71.45 | 66.82 | **68.71** | 78.58 | 91.17 | **87.78** | 51.01 | 50.16 | 70.57 | 53.86 | 70.90 | 1.82 |
| STAR-MMoE | **90.09** | **72.31** | **67.05** | 67.96 | **79.27** | **91.45** | 87.76 | **51.54** | **51.15** | **70.80** | **54.12** | **71.23** | **1.18** |
| | | | | | *STL10-pretrained ResNet-18* | | | | | | | | |
| SimCLR | 83.56 | 55.19 | 33.75 | 39.01 | 46.15 | 60.27 | 66.85 | 17.38 | 27.17 | 43.12 | 28.58 | 45.55 | 5.73 |
| AugSelf | 84.03 | 58.65 | 38.11 | 41.74 | 47.80 | 68.54 | 69.33 | 20.23 | 31.53 | 45.68 | 32.51 | 48.92 | 4.18 |
| EquiMod | 85.73 | 60.06 | 37.43 | 42.49 | 48.83 | 67.07 | 71.17 | 19.95 | 33.03 | 47.00 | 32.15 | 49.54 | 3.82 |
| CARE | 77.10 | 51.32 | **43.52** | **48.18** | 46.19 | 65.84 | 61.75 | 21.99 | 33.77 | **50.00** | 35.78 | 48.68 | 3.36 |
| STAR-SS | 85.64 | 61.10 | 40.17 | 44.50 | 50.07 | 72.59 | 73.39 | 21.89 | 33.90 | 48.58 | 34.25 | 51.46 | 2.55 |
| STAR-MMoE | **87.45** | **64.78** | 41.24 | 46.82 | **51.10** | **73.99** | **74.76** | **22.74** | **35.61** | 49.75 | 35.50 | **53.07** | **1.36** |

Table 2: **In-Domain Classification.** Linear evaluation accuracy (%) of ResNet-50 and ResNet-18 pretrained on ImageNet100 and STL10, respectively.

| Method | STL10 | ImageNet100 |
|---|---|---|
| SimCLR | 85.24 | 83.43 |
| AugSelf | 85.99 | 83.95 |
| EquiMod | **87.01** | 84.81 |
| CARE | 79.86 | 80.38 |
| STAR-SS | 86.75 | 84.82 |
| STAR-MMoE | 86.74 | **84.83** |

Table 3: **Object Detection.** Evaluation of learned representations on the object detection task using Faster R-CNN with a frozen ResNet-50-C4 backbone on VOC07+12. Reported metrics include AP, AP50, and AP75.

| Method | AP | AP50 | AP75 |
|---|---|---|---|
| SimCLR | 47.96±0.19 | 76.35±0.14 | 51.62±0.46 |
| AugSelf | 48.00±0.21 | 76.14±0.08 | 51.83±0.34 |
| EquiMod | 48.52±0.15 | 76.55±0.01 | 52.82±0.22 |
| CARE | 48.41±0.28 | 76.16±0.20 | 52.06±0.55 |
| STAR-SS | 48.77±0.16 | 76.64±0.03 | 52.77±0.31 |
| STAR-MMoE | **48.85±0.20** | **76.81±0.07** | **53.01±0.23** |

# 5 Experiments

**Setup.** We pretrain ResNet-18 [19] on STL10 [7] for 200 epochs and ResNet-50 on ImageNet100 [43, 45] for 500 epochs, both with a batch size of 256. For our proposed methods, we denote the single shared projection variant as **STAR-SS** and the MMoE projection variant as **STAR-MMoE**. We compare our methods against existing equivariant representation learning approaches, including AugSelf [27], EquiMod [9], and CARE [18]. For all approaches, SimCLR [3] is used as invariant representation learning baseline.

For STL10, we use 16 experts as the default configurations, while 8 experts are used for ImageNet100. For analysis, we use the STL10-pretrained model with 8 experts for better interpretability. All results are averaged over three runs, reporting mean and standard deviation, and we reproduce all compared methods to ensure fair comparisons. See Appendix B for detailed configurations.

## 5.1 Main Results

**Image Classification.** We conduct transfer learning experiments on 11 downstream datasets: CIFAR10/100 [26], Food [2], MIT67 [39], Pets [37], Flowers [34], Caltech101 [13], Cars [25], Aircraft [32], DTD [6], and SUN397 [49]. For evaluation, we follow the linear evaluation protocol used in [24]. Table 1 shows the transfer learning results across various downstream tasks. Out of the 11 downstream datasets, our method achieves the best performance in 7 when pretrained on STL10, and 10 when pretrained on ImageNet100, consistently surpassing previous approaches. As shown in Table 2, our method also achieves strong in-domain performance, suggesting that improvements in out-of-domain performance are achieved without degrading, or even potentially enhancing in-domain performance.

**Object Detection.** We evaluate our method on the object detection task using the Pascal VOC07+12 dataset [11]. According to [16], performance obtained after full fine-tuning reflects not only the quality of the learned representations, but also the effects of initialization and optimization strategies. Therefore, linear evaluation with a frozen backbone is more appropriate for assessing the quality of

Table 4: **Few-Shot Classification.** Few-shot classificiation accuracy (%) with 95% confidence intervals averaged over 2000 episodes. $(N, K)$ denotes $N$-way $K$-shot tasks. **Bold entries** indicate the best performance among methods, while underlined entries denote the second best.

| Method | FC100 | | CUB200 | | Plant Disease | |
|---|---|---|---|---|---|---|
| | $(5, 1)$ | $(5, 5)$ | $(5, 1)$ | $(5, 5)$ | $(5, 1)$ | $(5, 5)$ |
| *ImageNet100-pretrained ResNet-50* | | | | | | |
| SimCLR | 35.72±0.34 | 50.27±0.38 | 45.65±0.49 | 61.13±0.48 | 71.55±0.47 | 87.88±0.32 |
| AugSelf | 36.12±0.37 | 51.02±0.40 | 45.96±0.49 | 61.94±0.48 | 72.18±0.47 | 88.48±0.32 |
| EquiMod | 35.21±0.35 | 49.49±0.39 | 46.11±0.49 | 61.72±0.49 | 72.48±0.47 | 88.71±0.32 |
| CARE | 27.09±0.28 | 36.44±0.35 | 41.01±0.47 | 53.40±0.48 | 50.93±0.46 | 70.74±0.40 |
| STAR-SS | 35.80±0.37 | 51.33±0.39 | 47.11±0.49 | 62.12±0.47 | 72.48±0.48 | 89.46±0.29 |
| STAR-MMoE | **38.26±0.37** | **53.57±0.38** | **47.26±0.47** | **63.46±0.47** | **73.86±0.46** | **90.04±0.29** |
| *STL10-pretrained ResNet-18* | | | | | | |
| SimCLR | 36.49±0.35 | 51.46±0.44 | 35.55±0.36 | 50.18±0.37 | 56.89±0.48 | 75.86±0.39 |
| AugSelf | 37.99±0.37 | 52.81±0.39 | 38.34±0.38 | 54.02±0.39 | 59.76±0.49 | 78.31±0.36 |
| EquiMod | 36.42±0.36 | 50.64±0.36 | 39.08±0.43 | 52.88±0.44 | 58.25±0.48 | 78.01±0.37 |
| CARE | 32.93±0.39 | 43.04±0.40 | 36.93±0.42 | 51.28±0.43 | 58.62±0.49 | 79.67±0.36 |
| STAR-SS | 38.97±0.39 | 53.36±0.40 | 40.70±0.43 | 55.37±0.42 | 60.11±0.48 | 79.83±0.36 |
| STAR-MMoE | **39.35±0.39** | **54.87±0.40** | **41.46±0.42** | **57.31±0.45** | **61.70±0.47** | **81.14±0.36** |

the learned representations. Following this principle, we adopt a representation evaluation protocol analogous to that used in image classification, where the backbone is frozen to assess the quality of the learned representations. Specifically, we transfer pretrained weights to the Faster R-CNN [41] architecture with an R50-C4 backbone, freeze all convolutional layers from C1 to C4, and fine-tune only the region proposal network (RPN) and the object classification head (C5). As shown in Table 3, our method outperforms all baselines in AP, AP50, and AP75, demonstrating that it produces stronger and more transferable representations for object-level understanding.

**Few-Shot Classification.** We evaluate the generalizability of learned representations under limited data conditions via few-shot classification, following the linear evaluation protocol for few-shot learning in [27]. Specifically, we report classification accuracies and 95% confidence intervals for 5-way 1-shot and 5-way 5-shot tasks across 2000 episodes on FC100 [36], CUB200 [48], and Plant Disease datasets [33]. As shown in Table 4, our method consistently achieves the best performance across all few-shot classification settings, outperforming other equivariant representation learning methods in every case.

## 5.2 Analysis

**Shared Information between Invariant and Equivariant Learning.** To verify the existence of shared information between invariant and equivariant learning, we conduct $k$-NN retrieval on STL10 test set augmented with flipped images using invariance-only and equivariance-only models, where we employ SimCLR [3] as invariance-only model and EquiMod [9] without the invariant loss as equivariance-only model. Notably, the equivariance-only model retrieves semantically similar neighbors even without invariant learning. Specifically, in the results of the equivariance-only

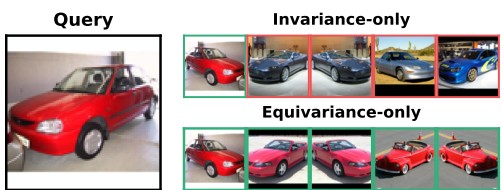

Figure 3: $k$-**NN Retrieval on STL10.** Query image (left); retrievals from the invariance-only model (top-right) and the equivariance-only model (bottom-right).

model, we observe three patterns: (i) it consistently retrieves samples belonging the same semantic object class, (ii) it consistently matches samples with the same color, and (iii) it alternates between samples of the same orientation with query and their flipped counterparts. These observations indicate that the equivariance-only model encodes not only augmentation-aware aspects such as orientation and color but also semantic cues like object identity, highlighting the shared information between invariant and equivariant learning.

**Expert Specialization.** We investigate how the MMoE projection module allocates experts to the invariant and equivariant objectives by analyzing the average routing weight distributions, measured

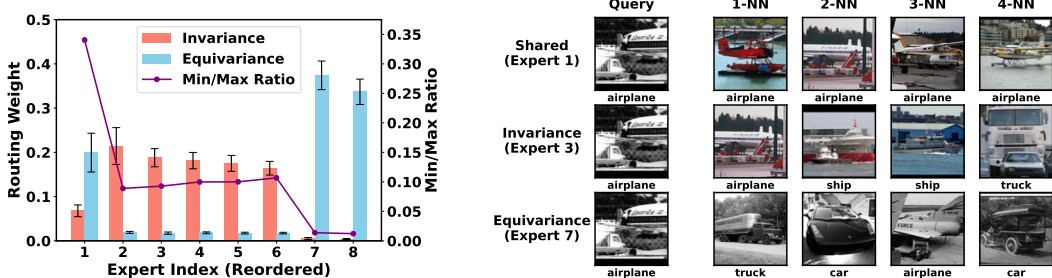

(a) Routing weight distribution across experts.

(b) $k$-NN retrieval on STL10 test dataset.

Figure 4: **Analysis of Expert Specialization.** (a) Routing weights averaged over test data in STL10, with experts reordered based on their roles. The min/max ratio (purple) measures the balance between how much each expert is utilized by the invariant and equivariant objectives. (b) $k$-NN retrieval results using the output embeddings of individual experts.

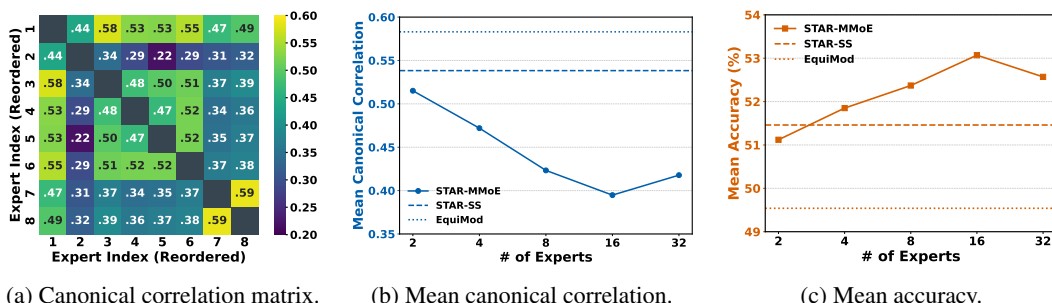

(a) Canonical correlation matrix.

(b) Mean canonical correlation.

(c) Mean accuracy.

Figure 5: **Analysis of Redundant Feature Learning.** (a) Pairwise canonical correlation between expert outputs. (b) Mean canonical correlation and (c) mean classification accuracy on 11 out-of-domain datasets across different numbers of experts in our proposed method. For (a), the numerical values of the diagonal elements are omitted for better visualization.

on the STL10 test dataset, as shown in Figure 4a. Expert 1 receives relatively balanced weights from both routers, implying that it learns information shared between the invariant and equivariant objectives. In contrast, Experts 2 to 6 are mainly used for the invariant objective, and Experts 7 and 8 are mainly used for the equivariant objective, indicating the task-specific specialization of experts. This distinction is further quantified by the min/max ratio, which computes the proportion of the smaller routing weight value to the larger one for each expert; higher values imply more balanced usage across tasks. To confirm these observations, we conduct $k$-NN retrieval on the STL10 test set, as shown in Figure 4b. Expert 3 and Expert 7 retrieve samples that emphasize invariance- and equivariance-specific features, respectively, while Expert 1 retrieves semantically consistent neighbors, indicating that it captures shared information beneficial to both objectives. These results demonstrate that the MMoE architecture supports meaningful expert specialization aligned with the learning objectives.

**Redundant Feature Learning.** We assess redundant feature learning among experts by measuring canonical correlation between their outputs, where a higher correlation indicates that experts capture similar, potentially redundant information. In Figure 5a, the matrix shows that experts assigned to the same objective (e.g., Experts 7 and 8) tend to exhibit higher mutual similarity, whereas the similarity between experts specialized in different objectives (e.g. Experts 3 and 7) remains relatively low. Notably, Expert 1 exhibits moderate similarity with both groups, indicating that it primarily encodes information shared across invariant and equivariant objectives.

To further quantify how redundant feature learning affects generalization, we vary the number of experts to control the degree of redundant feature learning in the model. As shown in Figure 5b, increasing the number of experts tends to reduce redundant feature learning, as reflected in the lower mean canonical correlation. This reduction in redundant feature learning is accompanied by consistent improvements in the mean accuracy, as shown in Figure 5c. Compared to baselines such as EquiMod, our method consistently achieves both reduced redundant feature learning and higher

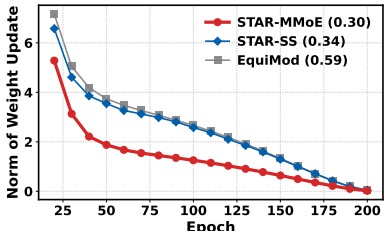

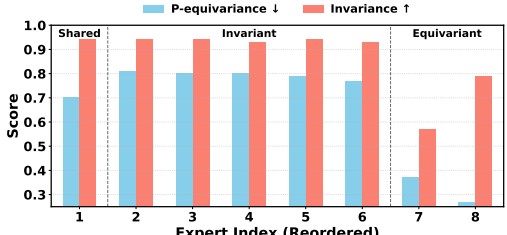

Figure 6: **Convergence of Experts.** Frobenius norm of weight update for experts over training epochs. All models are initialized and optimized using the same scheme to ensure a fair comparison of weight updates.

Figure 7: **Equivariance of Expert Embeddings.** P-equivariance and invariance of experts embeddings in our method pretrained on STL10.

accuracy, suggesting the effectiveness of dynamic expert allocation. These results highlight that reducing redundant feature learning through adaptive expert assignment plays a key role in promoting specialization and improving generalization.

**Impact of Redundant Feature Learning for Representations.**    To understand how redundant feature learning affects the backbone representations, we analyze its impact on expert convergence and the resulting gradient quality. As shown in Figure 6, experts in our method with the MMoE projection converge faster than those in EquiMod, consistent with prior findings that MoE architectures achieve faster convergence than dense models under the same number of training steps [28, 12, 40]. Since convergence of expert directly shapes the gradient signals to the backbone, faster convergence implies that our experts provide higher-quality and task-specific gradients to the backbone early in training, when the learning rate is high. In contrast, experts in EquiMod converge more slowly and yield suboptimal gradients, as redundant feature learning hinders task specialization and weakens task-specific updates. We empirically validate this by measuring the cosine similarity between gradients from different experts with respect to the backbone, which decreases from 0.59 in EquiMod to 0.30 in our method with MMoE projection. This result indicates reduced redundant feature learning and improved task specialization.

**Evaluation of Equivariance.**    We evaluate the equivariance of representations and experts using R-equivariance and P-equivariance, as proposed in [38]. R-equivariance, measured with cosine similarity, quantifies how well the transformed embedding can be predicted from the original embedding and the transformation parameters, whereas P-equivariance, measured with mean squared error, quantifies how accurately the transformation parameters can be recovered from the original and transformed embeddings. As shown in Table 5, representations from our method achieve higher R-equivariance and lower P-equivariance than those from other methods, indicating stronger equivariance.

Table 5: **Equivariance of Representations.** Comparison of R-equivariance and P-equivariance of representations learned by different methods pretrained on STL10.

| Method | R-equiv. ↑ | P-equiv. ↓ |
|---|---|---|
| SimCLR | 0.74 | 0.72 |
| AugSelf | 0.92 | 0.32 |
| EquiMod | 0.91 | 0.38 |
| CARE | 0.97 | 0.51 |
| STAR-SS | 0.93 | **0.27** |
| STAR-MMoE | **0.98** | **0.27** |

Figure 7 shows the P-equivariance and invariance scores across experts in our method. Invariance is measured as the cosine similarity between embeddings of differently augmented views. Equivariant experts exhibit lower P-equivariance and lower invariance, indicating that they specialize in the equivariant learning task and capture transformation-related information essential for learning equivariance. In contrast, invariant experts show higher P-equivariance and higher invariance, demonstrating they specialize in the invariant learning task. In addition, the shared expert shows moderate P-equivariance together with an invariance score comparable to that of invariant experts, reflecting its role in encoding shared information between invariant and equivariant learning. These results further confirm that our method achieves task specialization.

**Training Efficiency.**  Figure 8 compares the mean accuracy of different methods over training time in the out-of-domain classification setting, with wall-clock time measured on a single RTX 4090 GPU. While most methods steadily improve, SimCLR and AugSelf show saturation or even decline after 300 epochs, indicating a tendency to overfit to the training distribution and a limited ability to generalize to unseen domains. This observation is consistent with Figure A.1a, where their in-domain accuracy continues to increase beyond 300 epochs. In contrast, our proposed method consistently achieves higher accuracy and maintains this advantage throughout the entire training process. For example, our method surpasses other approaches trained for longer durations. Although our method may incur a higher cost per epoch, its ability to efficiently achieve strong generalization in less wall-clock time highlights its advantage in scenarios where early stopping is desirable.

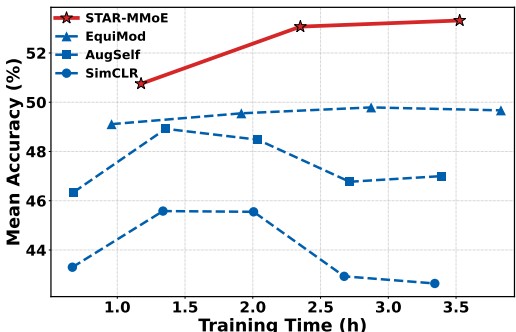

Figure 8: **Training Efficiency in Out-of-Domain Classification.** Mean accuracy (%) of STL10-pretrained models in out-of-domain classification. Markers are shown every 100 epochs.

# 6  Discussion and Conclusion

This work revisits equivariant representation learning and demonstrates that treating the invariant and equivariant objectives as fully independent, as commonly done in two-branch architectures, leads to redundant feature learning on experts. To address this, we propose STAR, which dynamically routes experts to each task. This reduces redundant feature learning and enhances generalization across downstream tasks such as image classification, few-shot learning, and object detection.

**Limitations.**  A limitation of our method is the use of soft routing, which prevents the use of sparse routing strategies such as top-$k$ routing. While sparse routing improves computational efficiency in conventional MoE models [44, 12], it causes unstable training in our setting due to the presence of batch normalization in each expert. Activating only a subset of experts per sample leads to unreliable batch statistics, especially problematic in SSL where stable statistics are essential. To avoid this, we adopt soft routing to ensure that all experts receive inputs in every batch. Although this reduces computational efficiency and scalability, it enables stable training dynamics.

## Acknowledgements

This work was partially supported by the National Research Foundation of Korea (NRF) grant funded by the Ministry of Science and ICT (MSIT) of the Korean government (RS-2024-00341749, RS-2024-00345351, RS-2024-00408003), and Institute of Information & Communications Technology Planning & Evaluation (IITP) grant funded by MSIT (RS-2023-00259934, RS-2025-02283048).

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

# Soft Task-Aware Routing of Experts for Equivariant Representation Learning
## Supplementary Material

## A  Additional Experiments

### A.1  Comparison with STL

Table A.1 compares the linear evaluation performance of our method against STL [51] and STL combined with AugMix [21] across 11 downstream classification benchmarks. We report results under two pretraining settings: STL10 with ResNet-18 and ImageNet100 with ResNet-50. Following the experimental setting of the STL paper, we evaluate on the AWS Flowers dataset, which differs from the Oxford Flowers [34] in its train/test split. Except for this replacement, all other evaluation protocols are kept identical.

Table A.1: **Image Classification.** Linear evaluation accuracy (%) across 11 datasets. Models are pretrained on STL10 using ResNet-18 or on ImageNet100 using ResNet-50. Symbol * denotes performance reported in [51], based on 200 epochs for STL10 and 500 epochs for ImageNet100. For Flowers, we use AWS Flowers instead of the Oxford Flowers.

| Pretraining | Method | In-domain | CIFAR10 | CIFAR100 | Food | MIT67 | Pets | Flowers | Caltech101 | Cars | Aircraft | DTD | SUN397 | Mean |
|---|---|---|---|---|---|---|---|---|---|---|---|---|---|---|
| ImageNet100 | STL* | 81.10 | 86.55 | 66.84 | 64.32 | 56.64 | 65.00 | 94.51 | 81.83 | 35.44 | 45.42 | 64.68 | 44.69 | 64.18 |
| | STL + AugMix* | 81.64 | 87.19 | 67.70 | 66.12 | 59.70 | 67.10 | **94.87** | 84.61 | 38.48 | 46.14 | 69.57 | 45.75 | 66.11 |
| | STAR-MMoE | **84.83** | **90.09** | **72.31** | **67.05** | **67.96** | **79.27** | 93.64 | **87.76** | **51.54** | **51.15** | **70.80** | **54.12** | **71.43** |
| STL10 | STL* | 84.83 | 85.22 | 60.13 | 38.05 | 43.53 | 46.57 | 73.50 | 71.36 | 18.85 | 30.25 | 45.34 | 31.63 | 49.49 |
| | STL + AugMix* | 85.57 | 86.01 | 62.07 | 40.16 | 44.90 | 46.69 | 77.37 | 73.29 | 19.32 | 30.87 | 48.71 | 33.44 | 51.17 |
| | STAR-MMoE | **86.74** | **87.45** | **64.78** | **41.24** | **46.82** | **51.10** | **78.92** | **74.76** | **22.74** | **35.61** | **49.75** | **35.50** | **53.52** |

Across both pretraining regimes, our method consistently outperforms STL and its AugMix variant on the majority of datasets. These results demonstrate the effectiveness of our method in producing more generalizable representations.

### A.2  Training Efficiency

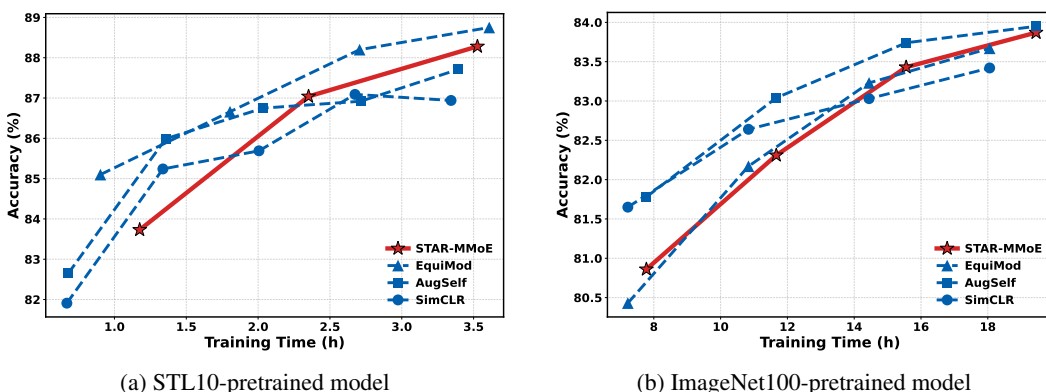

(a) STL10-pretrained model        (b) ImageNet100-pretrained model

Figure A.1: **Training Efficiency in In-Domain Classification.** Mean accuracy (%) of (a) STL10-pretrained and (b) ImageNet100-pretrained models in in-domain classification. For STL10 pretraining, all markers are shown every 100 epochs; for ImageNet100 pretraining, markers are shown every 100 epochs for SimCLR and AugSelf (from 200 epochs) and every 50 epochs for EquiMod and STAR-MMoE (from 100 epochs).

Figure A.1 shows that all methods follow similar convergence trends in the in-domain classification setting, with most reaching comparable accuracy levels as training progresses. Notably, STAR-MMoE achieves performance on par with AugSelf, the best method under ImageNet100 pretraining, and EquiMod, the best under STL10 pretraining.

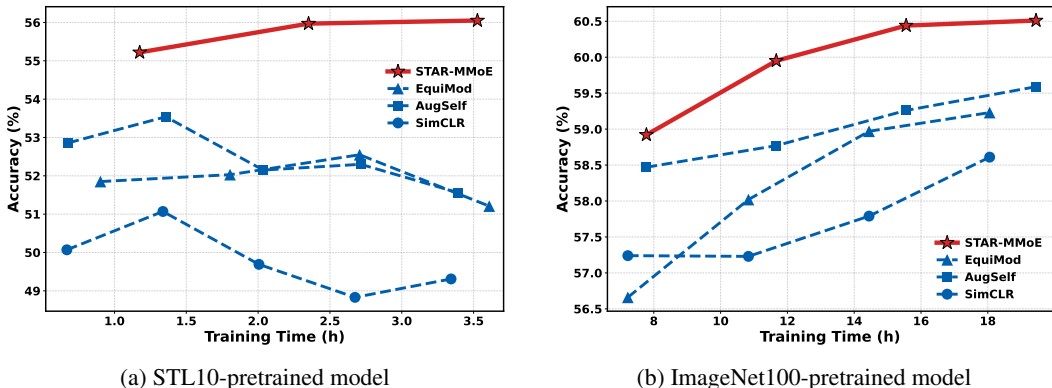

(a) STL10-pretrained model                    (b) ImageNet100-pretrained model

Figure A.2: **Training Efficiency in Few-Shot Classification.** Mean accuracy (%) of (a) STL10-pretrained and (b) ImageNet100-pretrained models in few-shot classification. For STL10 pretraining, all markers are shown every 100 epochs; for ImageNet100 pretraining, markers are shown every 100 epochs for SimCLR and AugSelf (from 200 epochs) and every 50 epochs for EquiMod and STAR-MMoE (from 100 epochs).

However, the distinction becomes more pronounced in the few-shot classification setting, as shown in Figure A.2. our method consistently outperforms all other methods, including those trained for longer durations. Figure A.2a further illustrates that, unlike STAR-MMoE, the methods pretrained with STL10 become saturated even with additional training.

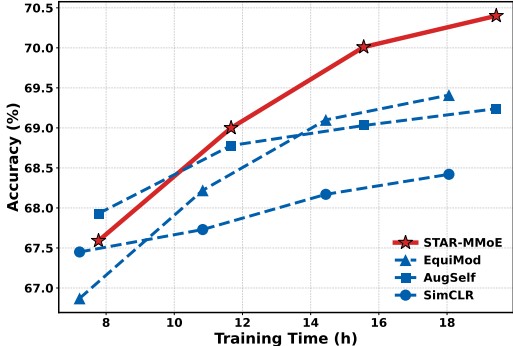

Figure A.3: **Training Efficiency in Out-of-Domain Classification.** Mean accuracy (%) of ImageNet100-pretrained models in out-of-domain classification. Markers are shown every 100 epochs for SimCLR and AugSelf (from 200 epochs) and every 50 epochs for EquiMod and STAR-MMoE (from 100 epochs).

Consistent with the results in Figure 8, Figure A.3 exhibits the same trend, further underscoring the strength of STAR-MMoE with ImageNet100 pretraining. In most cases, our method surpasses others even when they are trained for longer durations. This demonstrates that the proposed approach not only achieves superior transfer accuracy but also attains it more efficiently, making it particularly well suited for scenarios that demand strong generalization to unseen domains and high training efficiency.

## A.3 ViT Backbone

To assess the backbone independence of our proposed method, we conduct additional experiments using the Vision Transformer (ViT) [10] architecture. Specifically, we adopt MoCo-v3 [5], which utilizes ViT as its backbone, and modify SimCLR by replacing the original ResNet18 or ResNet50

with ViT. Following the experimental setup introduced in [35], ViT-Small is pretrained on ImageNet100 for 200 epochs with a batch size of 256. The training setup commonly employs the AdamW optimizer [30], with a linear warm-up of the learning rate during the first 40 epochs, a momentum of 0.9, and a weight decay of 0.1. A cosine learning rate schedule [29] is used for both the encoder and the projector, while in the case of MoCo-v3, the same schedule is also applied to the predictor.

**MoCo-v3.** We adopt the original parameter settings, with a learning rate of 1.5e-4 and a temperature of 0.2 for both contrastive and equivariant learning. The exponential moving average (EMA) coefficient is initialized at 0.99 and gradually increased to 1. We use a 3-layer MLP for each expert, with hidden and output dimensions set 4096 and 256, respectively. The equivariant predictor is a single-layer MLP with an input dimension of 512 and an output dimension of 256.

**SimCLR.** The learning rate is set to 1.5e-3, and the temperature for contrastive and equivariant learning is also fixed at 0.2. We use a 3-layer MLP for each expert, with hidden and output dimensions set to 4096 and 256, respectively. The equivariant predictor is a single-layer MLP with an input dimension of 512 and an output dimension of 256.

Table A.2: **Backbone Ablation Study.** Linear evaluation accuracy (%) of ViT-S/16 pretrained on ImageNet100.

| Baseline | Method | CIFAR10 | CIFAR100 | Food | MIT67 | Pets | Flowers | Caltech101 | Cars | Aircraft | DTD | SUN397 | Mean |
|---|---|---|---|---|---|---|---|---|---|---|---|---|---|
| SimCLR | - | 85.19±0.41 | 65.17±0.10 | 56.67±0.19 | 57.21±1.37 | 66.24±0.42 | 84.89±0.22 | 75.04±0.18 | 30.67±0.46 | 35.66±0.33 | 61.15±0.96 | 44.91±0.22 | 60.25 |
| | AugSelf | 85.69±0.34 | 65.88±0.44 | 57.36±0.22 | 57.39±0.31 | 66.89±0.18 | 84.89±0.26 | 75.38±0.39 | 30.67±0.32 | 36.00±0.14 | 61.44±1.13 | 45.42±0.18 | 60.64 |
| | EquiMod | 86.81±0.28 | 67.47±0.50 | 59.34±0.14 | **60.40±0.37** | 69.86±0.86 | 86.90±0.32 | 78.12±0.15 | 33.71±0.52 | 38.67±0.27 | 62.87±0.52 | **47.08±0.03** | 62.84 |
| | Ours | **86.95±0.21** | **68.24±0.18** | **59.73±0.12** | 59.75±0.59 | **68.49±0.20** | **87.43±0.40** | **79.42±0.75** | **35.96±0.70** | **39.96±0.26** | **62.91±0.18** | 46.84±0.23 | **63.24** |
| MoCo-v3 | - | 84.96±0.23 | 64.85±0.20 | 57.74±0.04 | 57.74±1.32 | 65.99±0.48 | 84.69±0.38 | 75.85±0.27 | 30.45±0.39 | 35.91±0.37 | 60.89±0.57 | 45.48±0.15 | 60.41 |
| | AugSelf | 85.83±0.30 | 66.41±0.24 | 58.66±0.23 | 58.21±0.32 | 66.00±0.46 | 85.57±0.17 | 76.58±0.16 | 30.57±0.72 | 36.12±0.42 | 60.67±0.48 | 45.71±0.23 | 60.94 |
| | EquiMod | 85.98±0.17 | 66.52±0.38 | 59.69±0.26 | **59.75±0.85** | **67.88±0.35** | 87.30±0.17 | 78.01±0.76 | 32.01±0.58 | 37.76±0.49 | **63.21±0.24** | 46.82±0.11 | 62.23 |
| | Ours | **86.72±0.08** | **67.89±0.20** | **60.38±0.24** | 60.40±0.19 | 67.57±0.30 | **87.84±0.22** | **79.64±0.32** | **35.27±0.32** | **39.20±0.53** | 62.93±1.18 | **47.73±0.16** | 63.23 |

In Table A.2, our method achieves better performance than existing equivariant representation learning approaches across most datasets. This highlights the effectiveness of the proposed MMoE projection module, which is applicable to the ViT backbone.

## A.4 Ablation Study on Components

We conduct an ablation study to assess the contribution of each component in our method. As shown in Table A.3, substituting the projection head in EquiMod with the MMoE projection module leads to clear improvements in both in- and out-of-domain settings, highlighting its effectiveness in reducing redundant feature learning through dynamic expert allocation. In both EquiMod and our proposed method, adding the residual connection (RC) and increasing the depth of the equivariant predictor (PD) contribute to improved out-of-domain generalization. However, the deeper predictor slightly reduces the in-domain performance. Combining all three components results in the best out-of-domain performance, demonstrating their complementary contributions to both expressivity and generalization.

Table A.3: **Ablation Study on Components. MMoE**: MMoE projection module; **RC**: residual connection; **PD**: deeper equivariant predictor.

| Method | MMoE | RC | PD | In-domain | Out-domain |
|---|---|---|---|---|---|
| EquiMod | ✗ | ✗ | ✗ | 87.01 | 49.54 |
| | ✗ | ✓ | ✗ | 87.19 | 49.78 |
| | ✗ | ✗ | ✓ | 86.03 | 50.85 |
| | ✗ | ✓ | ✓ | 87.01 | 50.99 |
| STAR-MMoE | ✓ | ✗ | ✗ | 87.39 | 51.43 |
| | ✓ | ✓ | ✗ | **87.41** | 52.07 |
| | ✓ | ✗ | ✓ | 86.57 | 52.53 |
| | ✓ | ✓ | ✓ | 86.74 | **53.07** |

## A.5 Ablation Study on Baseline Invariant Learning Methods

Although the main results are presented based on SimCLR, as shown in Table A.4, our method generalizes well when combined with other invariant learning objectives, consistently outperforming all methods across various baseline invariant learning methods, including MoCo, SimSiam, and BYOL. Notably, EquiMod performs worse than AugSelf in non-contrastive frameworks such as SimSiam and BYOL, whereas our method maintains strong performance across both contrastive and non-contrastive frameworks.

Table A.4: **Ablation Study on Baseline Invariant Learning Methods.** Linear evaluation accuracy (%) of ResNet-18 pretrained on STL10 with various methods across SSL frameworks.

| Baseline | Method | STL10 | CIFAR10 | CIFAR100 | Food | MIT67 | Pets | Flowers | Caltech101 | Cars | Aircraft | DTD | SUN397 | Mean |
|---|---|---|---|---|---|---|---|---|---|---|---|---|---|---|
| MoCo | - | 81.98±0.49 | 83.69±0.48 | 58.02±0.67 | 34.74±0.23 | 40.10±0.71 | 42.13±0.24 | 63.64±0.22 | 65.78±0.13 | 16.87±0.40 | 28.93±0.66 | 43.53±0.65 | 29.89±0.28 | 46.12 |
| | AugSelf | 82.27±0.31 | 84.49±0.20 | 60.29±0.34 | 37.68±0.19 | 42.12±1.29 | 45.05±0.33 | 68.15±0.12 | 66.95±0.36 | 17.98±0.06 | 30.65±0.82 | 45.41±0.29 | 31.79±0.16 | 48.23 |
| | EquiMod | 85.92±0.20 | 86.44±0.14 | 62.28±0.32 | 38.67±0.21 | 44.18±0.50 | 47.46±0.22 | 69.88±0.27 | 70.42±0.44 | 19.56±0.36 | 32.77±1.39 | 46.86±0.28 | 33.07±0.25 | 50.15 |
| | STAR-MMoE | 86.93±0.07 | 87.40±0.22 | 63.89±0.28 | 39.76±0.21 | 45.80±0.49 | 49.49±0.28 | 72.25±0.13 | 72.62±0.52 | 21.35±0.39 | 33.89±0.47 | 48.23±0.56 | 34.33±0.16 | 51.73 |
| SimSiam | - | 85.46±0.10 | 82.48±0.66 | 54.43±0.90 | 34.38±0.18 | 39.65±0.65 | 45.68±0.29 | 58.93±0.42 | 66.62±1.05 | 17.28±0.16 | 27.23±0.99 | 42.78±0.99 | 28.83±0.10 | 45.30 |
| | AugSelf | 86.06±0.13 | 86.20±0.23 | 62.97±0.24 | 41.71±0.05 | 44.90±0.63 | 49.59±0.51 | 73.08±0.12 | 72.47±0.93 | 21.24±0.54 | 33.82±0.42 | 48.10±0.24 | 34.49±0.08 | 51.69 |
| | EquiMod | 87.05±0.35 | 85.94±0.99 | 61.02±1.33 | 39.39±0.37 | 43.83±0.41 | 50.09±0.30 | 69.61±0.54 | 71.48±0.72 | 20.41±0.50 | 33.02±0.39 | 47.75±1.26 | 32.94±0.03 | 50.50 |
| | STAR-MMoE | 87.85±0.31 | 85.84±0.72 | 62.33±1.29 | 41.36±0.11 | 46.39±0.80 | 51.35±0.11 | 73.37±0.49 | 73.99±0.54 | 22.88±0.45 | 35.56±0.66 | 48.88±0.37 | 34.67±0.17 | 52.42 |
| BYOL | - | 87.10±0.13 | 86.39±0.16 | 60.89±0.19 | 37.38±0.18 | 41.79±0.53 | 50.95±0.21 | 67.42±0.55 | 70.43±1.38 | 23.70±0.71 | 32.18±0.71 | 44.79±0.81 | 31.69±0.05 | 49.78 |
| | AugSelf | 87.13±0.46 | 86.95±0.20 | 64.34±0.15 | 43.38±0.18 | 45.62±0.29 | 52.73±0.60 | 74.45±0.43 | 73.86±0.21 | 25.56±0.55 | 35.50±0.54 | 49.04±0.09 | 34.96±0.26 | 53.31 |
| | EquiMod | 87.70±0.20 | 86.75±0.11 | 62.88±0.17 | 41.22±0.24 | 45.55±0.17 | 51.60±0.21 | 73.96±0.26 | 74.40±0.38 | 23.75±0.46 | 36.46±0.51 | 49.61±0.38 | 35.40±0.09 | 52.87 |
| | STAR-MMoE | 86.44±0.39 | 86.70±0.30 | 64.25±0.21 | 43.25±0.37 | 47.21±0.63 | 52.45±0.69 | 76.22±0.31 | 74.76±0.33 | 24.86±0.66 | 38.07±0.80 | 49.80±0.42 | 36.21±0.32 | 53.98 |

## A.6 Hyperparameter Optimization

We study how variation of hyperparameters can influence our model. For that purpose, we train ResNet-18 on STL10 with 16 experts and ResNet-50 on ImageNet100 for each factor modification, and present results for both in-domain and out-domain scenarios. We mainly inspect the influence of $\lambda$, the balancing coefficient of the equivariant loss (see Eq. (6)), and $\tau$, the temperature parameter used in the equivariant learning objective (see Eq. (14)). All hyperparameters are tuned based on out-of-domain performance to ensure generalization. As shown in Table A.5, both parameters have a notable impact on performance.

Table A.5: **Hyperparameter Tuning for $\lambda$ and $\tau$.** $\lambda$ is the loss balancing coefficient, and $\tau$ is the temperature parameter used in the equivariant learning objective. We report performance on in-domain and out-domain settings under varying values of each.

| Pretraining | $\lambda$ | In-domain | Out-of-domain |
|---|---|---|---|
| | 0.1 | 84.02 | 70.32 |
| | 0.2 | 84.20 | 70.82 |
| | 0.5 | 84.66 | 70.94 |
| ImageNet100 | 1 | **84.83** | **71.23** |
| | 2 | 84.79 | 70.61 |
| | 5 | 84.68 | 70.44 |
| | 10 | 83.82 | 69.14 |
| | 0.1 | 86.47 | 50.37 |
| | 0.2 | 86.46 | 51.27 |
| | 0.5 | 86.65 | 52.25 |
| STL10 | 1 | **86.74** | **53.07** |
| | 2 | 86.17 | 52.52 |
| | 5 | 84.95 | 50.88 |
| | 10 | 82.64 | 48.87 |

(a) $\lambda$: Loss balancing coefficient

| Pretraining | $d$ | In-domain | Out-of-domain |
|---|---|---|---|
| | 0.05 | 84.12 | 70.53 |
| | 0.1 | 84.70 | 70.86 |
| ImageNet100 | 0.2 | **84.83** | **71.23** |
| | 0.5 | 84.20 | 70.98 |
| | 1 | 84.10 | 70.74 |
| | 0.05 | 85.84 | 51.74 |
| | 0.1 | **86.98** | 52.59 |
| STL10 | 0.2 | 86.74 | **53.07** |
| | 0.5 | 86.66 | 52.09 |
| | 1 | 86.45 | 50.66 |

(b) $\tau$: Equivariant temperature parameter

We observe that $\lambda$ controls the relative strength of the equivariant learning signal. Increasing $\lambda$ improves performance up to a certain point, with the best results observed at $\lambda = 1$, beyond which performance drops due to the overemphasis on equivariance. This highlights the importance of balancing equivariant and invariant objectives to prevent one from dominating the learning process. For $\tau$, which modulates the sharpness of the similarity distribution in the equivariant contrastive loss, we find that $\tau = 0.2$ achieves optimal results. Smaller values such as 0.05 may cause representational

collapse due to overly confident similarity distributions, whereas larger values such as 1 reduce the discriminative power of the model. These trends are consistent across both in-domain and out-domain evaluations, emphasizing the necessity of careful calibration of hyperparameters for generalizable representation learning. Moreover, the similar tendencies for both $\lambda$ and $\tau$ are observed across STL10 and ImageNet100 pretraining settings, and we find that EquiMod exhibits the same optimal hyperparameter configuration, indicating that these settings generalize well across related architectures.

## A.7 Qualitative Analysis of Learned Representations

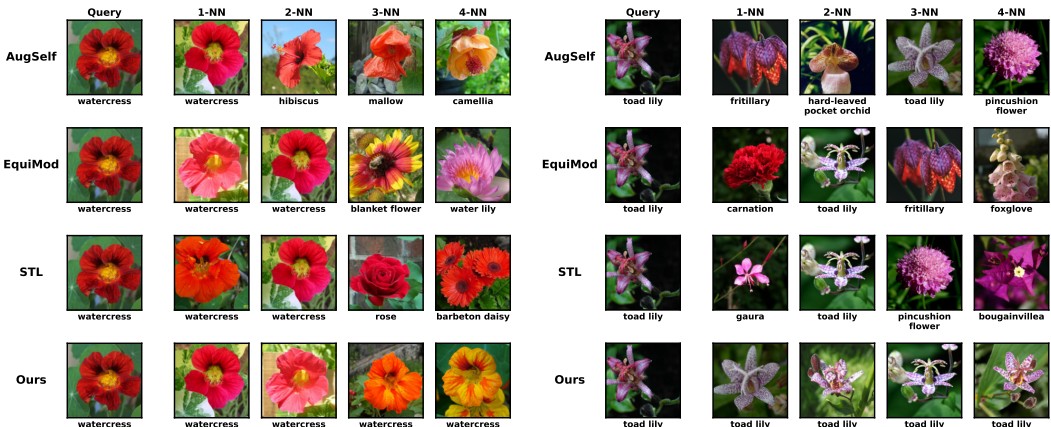

Figure A.4: $k$-**NN Retrieval on Flowers Test Set.** Results of $k$-NN retrieval using backbone features learned by Equivariant SSL methods.

Figure A.4 presents qualitative results of $k$-NN retrieval on the Flowers test set using backbone features from equivariant SSL methods. All models are based on ResNet-18 pretrained on STL10. We observe that most methods exhibit strong sensitivity to color information, often retrieving visually similar but semantically incorrect samples. For example, methods such as AugSelf and EquiMod frequently return instances from different classes that share similar color distributions with the query. STL also tends to favor samples with matching low-level visual cues, rather than consistently retrieving semantically correct instances. This indicates that their learned representations may not fully capture semantic object-level features.

In contrast, our method consistently retrieves samples from the same class as the query while still preserving sensitivity to fine-grained visual details such as color and texture. This suggests that our proposed method enables the model to encode features that are consistent with the class more effectively. Quantitatively, this advantage is reflected in a performance improvement of approximately 5 percentage points compared to the method that performs best on the Flowers dataset.

## A.8 Retrieval Visualization of Individual Experts

Figure A.5 provides full $k$-NN retrieval results using the output embeddings of all individual experts. While Figure 4b in the main paper focuses on three specific experts, namely the shared expert (Expert 1), the invariant expert (Expert 3), and the equivariant expert (Expert 7), this extended visualization enables a more comprehensive examination of the specialization exhibited by all experts.

We observe diverse retrieval patterns across experts. Expert 1 consistently retrieves semantically similar instances, indicating a focus on object-level meaning. Experts 2 through 6 often retrieve samples that differ in color from the query, suggesting that these experts have learned to ignore color and instead emphasize more abstract features. In contrast, Expert 7 and Expert 8 tend to retrieve samples with similar color characteristics, indicating that these experts have learned to capture information related to the augmentations.

This qualitative evidence supports the notion that the MMoE architecture induces meaningful division of roles among experts, with some specializing in robust semantic consistency and others in transformation sensitivity. Such functional diversity contributes to the model's capacity to generalize across tasks and domains.

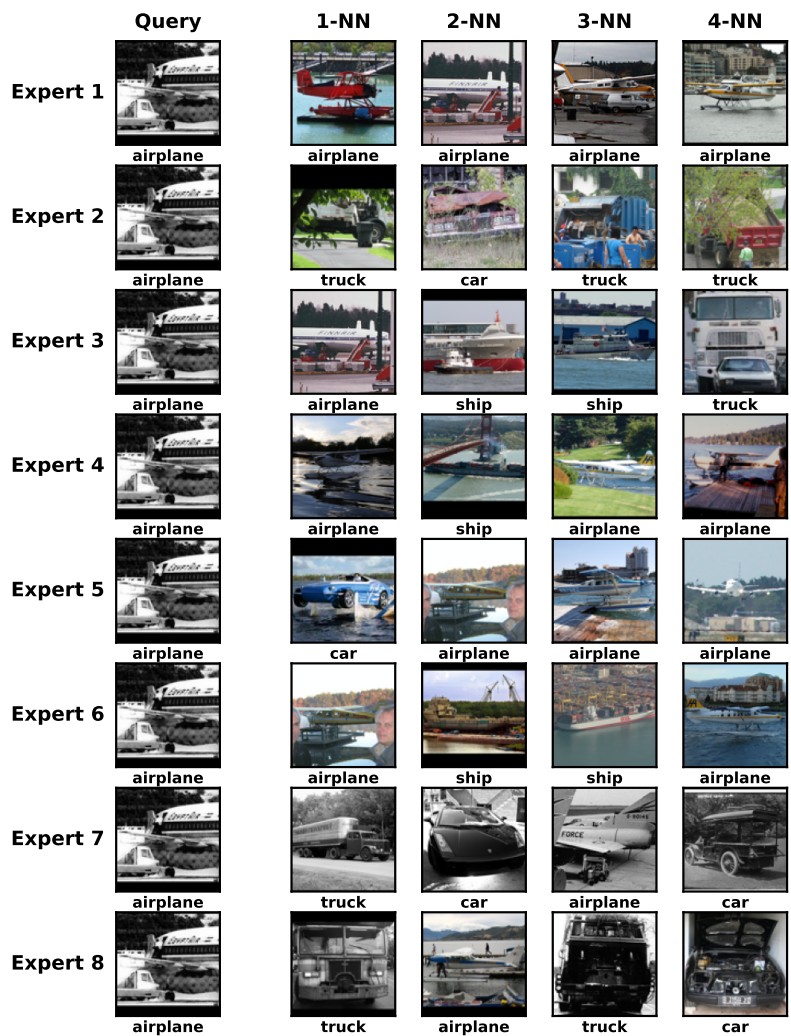

Figure A.5: *k*-**NN Retrieval on STL10 Test Set.** *k*-NN retrieval results using the output embeddings of all individual experts.

## A.9 Expert Specialization in ImageNet100 Pretraining

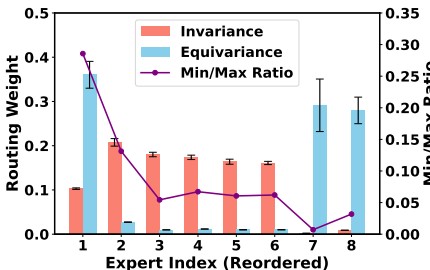

Figure A.6: **Routing Weight Distribution Across Experts.** Routing weights averaged over test data in ImageNet100, with experts reordered based on their roles.

We investigate expert specialization in ImageNet100 pretraining following the same procedure as in Figure 4. Figure A.6 illustrates that Expert 1 receives relatively balanced routing weights from both invariant and equivariant branches, whereas Experts 2–6 are predominantly used for invariant learning and Experts 7–8 for equivariant learning. This observation indicates that the experts are well

specialized for their respective learning objectives. Furthermore, Figure A.7 presents $k$-NN retrieval results on a subset of the ImageNet100 test set, where we randomly select 10 classes for visualization. Consistent with Figure A.5, each expert retrieves samples aligned with its designated role, confirming that our method effectively promotes expert specialization in the ImageNet100 pretraining setting as well.

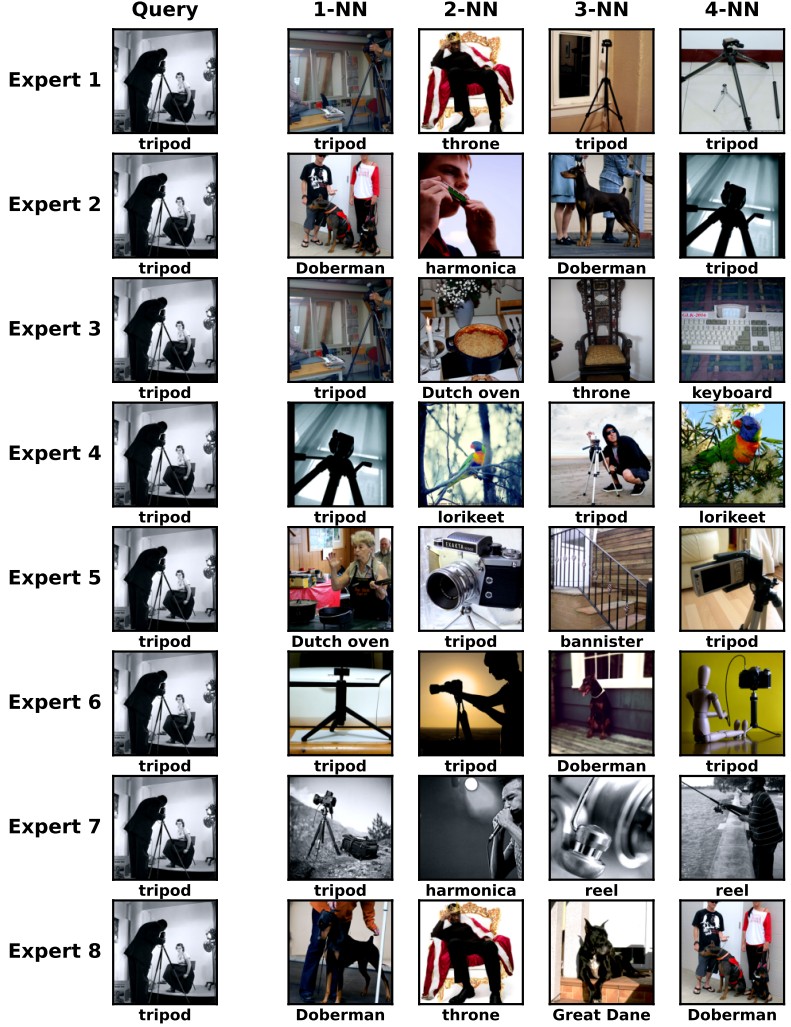

Figure A.7: $k$-**NN Retrieval on ImageNet100 Test Set.** $k$-NN retrieval results using the output embeddings of all individual experts.

# B  Experimental Setup

## B.1  Datasets

Table B.1 summarizes the datasets used in our experiments. Category (a) includes the pretraining datasets, STL10 and ImageNet100, which are exclusively used during the unsupervised pretraining phase. Category (b) covers the datasets used for linear evaluation, with each dataset annotated by the number of classes and the number of samples in the training, validation, and test splits. For datasets without an official validation split, validation samples are randomly selected from the training set. Category (c) consists of few-shot benchmarks, including the meta-test split of FC100 [36], as well as the full datasets of CUB200 [48] and Plant Disease [33]. Finally, Category (d) comprises object detection datasets, where we use the `trainval` split of VOC07+12 [11] for training and the `test` split for evaluation.

Table B.1: **Dataset Information.** Overview of the datasets used in the experiments. This table lists dataset names, the number of classes, and the counts for training, validation, and test samples, along with the evaluation metrics.

| Category | Dataset | # of classes | Training | Validation | Test | Metric |
|---|---|---|---|---|---|---|
| (a) Pretraining | STL10 [7] | 10 | 105,000 | - | - | - |
| | ImageNet100 [43, 45] | 1000 | 126,689 | - | - | - |
| (b) Linear Evaluation | CIFAR10 [26] | 10 | 45,000 | 5,000 | 10,000 | Top-1 accuracy |
| | CIFAR100 [26] | 100 | 45,000 | 5,000 | 10,000 | Top-1 accuracy |
| | Food [2] | 101 | 68,175 | 7,575 | 25,250 | Top-1 accuracy |
| | MIT67 [39] | 67 | 4,690 | 670 | 1,340 | Top-1 accuracy |
| | Pets [37] | 37 | 2,940 | 740 | 3,669 | Mean per-class accuracy |
| | Flowers [34] | 102 | 1,020 | 1,020 | 6,149 | Mean per-class accuracy |
| | Caltech101 [13] | 101 | 2,525 | 505 | 5,647 | Mean per-class accuracy |
| | Cars [25] | 196 | 6,494 | 1,650 | 8,041 | Top-1 accuracy |
| | Aircraft [32] | 100 | 3,334 | 3,333 | 3,333 | Mean per-class accuracy |
| | DTD (split 1) [6] | 47 | 1,880 | 1,880 | 1,880 | Top-1 accuracy |
| | SUN397 (split 1) [49] | 397 | 15,880 | 3,970 | 19,850 | Top-1 accuracy |
| (c) Few-shot | FC100 [36] | 20 | - | - | 12,000 | Average accuracy |
| | CUB200 [48] | 200 | - | - | 11,780 | Average accuracy |
| | Plant Disease [33] | 38 | - | - | 54,305 | Average accuracy |
| (d) Object Detection | VOC2007+2012 [11] | 20 | 16,551 | - | 4,952 | Average precision |

## B.2 Pretraining Setups

### B.2.1 ImageNet100 Pretraining

We pretrain a ResNet-50 backbone [19] on ImageNet100, a 100-class subset of ImageNet [43], following the dataset splits in [45]. The model is trained using the SimCLR framework [3], employing stochastic gradient descent (SGD) for 500 epochs with a batch size of 256. A cosine annealing learning rate schedule [29] is used, initialized at 0.03 and without restarts, and a weight decay of 0.0005 is applied. The architecture includes 8 experts $\{E_i\}_{i=1}^{8}$, each implemented as a 3-layer MLP with a hidden dimension of 2048 and an output dimension of 128. Batch normalization [22] is excluded from the final layer of each expert. The equivariant predictor $\phi_T$ consists of 3 layers, each with a hidden dimension of 512. The routers $R^{\text{inv}}$ and $R^{\text{eq}}$ are implemented as single-layer MLPs that output 8-dimensional vectors corresponding to the number of experts, followed by softmax activations to produce normalized weights over experts. Pretraining on ImageNet100 is performed using 4 NVIDIA RTX 4090 GPUs.

### B.2.2 STL10 Pretraining

We pretrain a ResNet-18 backbone on STL10 using stochastic gradient descent (SGD). Training is conducted for 200 epochs with a batch size of 256. A cosine annealing learning rate schedule without restarts is used, with the initial learning rate set to 0.03, except for SimSiam where a learning rate of 0.05 is used. A weight decay of 0.0005 is applied. The routers are implemented in the same manner as those used for ImageNet100 pretraining. For STL10, pretraining is conducted on a single NVIDIA RTX 4090 GPU.

**SimCLR.** We use a 3-layer expert architecture with 16 experts $\{E_i\}_{i=1}^{16}$, each with hidden and output dimensions of 512 and 128, respectively. Batch normalization is excluded from the final layer of each expert. The equivariant predictor is a 3-layer MLP with a hidden dimension of 512. A temperature parameter of 0.2 is used consistently for both contrastive and equivariant learning objectives.

**MoCo.** A 3-layer expert architecture with 8 experts is employed, where each expert has a hidden dimension of 512 and an output dimension of 128. Batch normalization is excluded from the final layer. The equivariant predictor is a single-layer MLP. A temperature parameter of 0.2 is used consistently for both contrastive and equivariant learning objectives.

**SimSiam.** We use a 2-layer expert architecture with 4 experts, each having hidden and output dimensions of 2048. Batch normalization is excluded from the final layer. The equivariant predictor is a single-layer MLP with input and output dimensions of 2048. A temperature of 0.1 is used for equivariant learning.

**BYOL.** A 2-layer expert architecture is used, consisting of 4 experts with a hidden dimension of 4096 and an output dimension of 256. Batch normalization is excluded from the final layer. The equivariant predictor is a 2-layer MLP with a hidden dimension of 512. A temperature of 0.1 is used for equivariant learning.

## B.3 Evaluation Protocol

**Linear Evaluation.** We adopt the standard linear evaluation protocol [3, 17, 24], where a linear classifier is trained on top of frozen features extracted from center-cropped images of size $224 \times 224$ (or $96 \times 96$ when pretrained on STL10), without any data augmentation. Specifically, each image is first resized so that its shorter side is 224 pixels, followed by a center crop of size $224 \times 224$. The classifier is optimized using an $\ell_2$-regularized cross-entropy objective with L-BFGS. The regularization strength is selected based on validation accuracy from 45 logarithmically spaced values ranging from $10^{-6}$ to $10^5$, and the final test accuracy is reported using the best model. We set the maximum number of L-BFGS iterations to 5000 and employ warm-start initialization by using the previous solution as the starting point for the next optimization step.

**Few-Shot Classification.** To evaluate representations in few-shot benchmarks, we perform logistic regression on top of frozen features using $N \times K$ support samples, without any fine-tuning or data augmentation, within each $N$-way $K$-shot episode.

**Object Detection.** We train a Faster R-CNN [41] with a R50-C4 backbone on the VOC2007+2012 `trainval` split containing 16551 images. To assess the quality of learned representations, we freeze all convolutional layers from C1 to C4 and train only the region proposal network (RPN) and the object classification head C5. The model is optimized for 24000 iterations with a batch size of 16 using synchronized batch normalization. The learning rate is set to 0.1 initially and decays by a factor of 10 at 18000 and 22000 iterations. A linear warmup [15] is applied during the first 1000 iterations with slope 0.333.

**R-equivariance and P-equivariance.** To evaluate R-equivariance, we train a linear layer to predict the embedding of an augmented image from the original image embedding and its corresponding augmentation parameters. The augmentation parameters are first projected to a 32-dimensional vector using a single-layer projector. We then concatenate the projected augmentation parameters with the original image embedding and feed the result into a one-layer predictor to generate the predicted embedding of augmented image. Finally, we compute the cosine similarity between the predicted and ground-truth embedding of augmented image.

For P-equivariance, we train a 1-layer predictor to estimate the augmentation parameters from the embeddings of the original and augmented images. Specifically, we concatenate the embeddings of the original and augmented images and feed the resulting vector into the predictor. Finally, we compute the mean-squared error (MSE) between the predicted and ground-truth augmentation parameters.

## B.4 Augmentations

In this section, we describe how augmentation parameters are defined based on the transformations used in AugSelf [27] including random crop, horizontal flip, color jitter, grayscale, and Gaussian blur. Each parameter set is defined according to the specific configuration of each transformation. In our method, all parameters are normalized using the empirical mean and standard deviation of each transformation-specific variable before being projected into the embedding space. These normalized parameters are then projected into the same dimensional space as the equivariant embedding through a single linear layer.

- `RandomResizedCrop`. The parameter is defined by the center coordinates $H_{\text{center}}$ and $W_{\text{center}}$ of the crop, along with the crop size given by height $H$ and width $W$. The crop is applied to images resized to 96×96 for STL10 and 224×224 for ImageNet100.

- `RandomHorizontalFlip`. This transformation is applied with a probability of 0.5. Since the operation is binary, the parameter is defined as either 0 or 1.

- `ColorJitter`. Color jitter includes four parameters: brightness, contrast, saturation, and hue. It is applied with a probability of 0.8. The maximum strength is set to 0.4 for brightness, contrast, and saturation, and 0.1 for hue. Each parameter is sampled independently from the ranges [0.6, 1.4] for

brightness, contrast, and saturation, and $[-0.1, 0.1]$ for hue. The transformations are applied in a random order rather than a fixed sequence. If `ColorJitter` is not applied, a default parameter of $[1, 1, 1, 0]$ is used.

- `RandomGrayScale.` Grayscale conversion is applied with a probability of 0.2. Similar to flipping, the parameter is binary with values of 0 or 1.

- `GaussianBlur.` The parameter consists of both the standard deviation of the blur, which ranges from 0.1 to 2.0, and a binary flag indicating whether the transformation was applied.

Table B.2: **Out-of-Domain Classification.** Linear evaluation accuracy (%) of ResNet-50 and ResNet-18 pretrained on ImageNet100 and STL10, respectively. **Bold entries** indicate the best performance among methods, while underlined entries denote the second best.

| Method | CIFAR10 | CIFAR100 | Food | MIT67 | Pets | Flowers | Caltech101 | Cars | Aircraft | DTD | SUN397 | Mean | Avg. Rank |
|---|---|---|---|---|---|---|---|---|---|---|---|---|---|
| *ImageNet100-pretrained ResNet-50* | | | | | | | | | | | | | |
| SimCLR | 87.88±0.27 | 67.92±0.19 | 63.60±0.32 | 66.57±1.0 | 76.71±0.77 | 88.37±0.37 | 85.02±0.20 | 47.09±0.21 | 48.23±0.36 | 69.17±0.28 | 52.02±0.27 | 68.42±0.20 | 5.00 |
| AugSelf | 88.61±0.12 | 69.68±0.16 | 65.37±0.22 | 67.51±0.43 | 77.24±0.38 | 89.70±0.32 | 85.09±0.14 | 47.48±0.44 | 48.65±0.28 | 69.31±0.83 | 53.00±0.12 | 69.24±0.22 | 3.82 |
| EquiMod | 88.99±0.18 | 70.22±0.20 | 64.43±0.15 | 67.54±0.43 | 77.78±0.13 | 90.33±0.02 | 86.62±0.09 | 48.94±0.20 | 49.91±0.42 | 69.33±0.17 | 52.79±0.23 | 69.72±0.17 | 3.18 |
| CARE | 82.81±0.19 | 58.97±0.43 | 55.78±0.10 | 56.39±0.37 | 59.89±0.80 | 75.84±0.14 | 73.14±0.16 | 29.12±0.59 | 36.00±0.48 | 62.22±0.81 | 42.48±0.25 | 57.51±0.42 | 6.00 |
| STAR-SS | 89.81±0.06 | 71.45±0.28 | 66.82±0.06 | **68.71±0.84** | 78.58±0.42 | 91.17±0.22 | **87.78±0.53** | 51.01±0.60 | 50.16±0.25 | 70.57±0.15 | 53.86±0.15 | 70.90±0.29 | 1.82 |
| STAR-MMoE | **90.09±0.12** | **72.31±0.27** | **67.05±0.11** | 67.96±0.47 | **79.27±0.27** | **91.45±0.20** | 87.76±0.25 | **51.54±0.56** | **51.15±0.49** | **70.80±0.20** | **54.12±0.10** | **71.23±0.21** | **1.18** |
| *STL10-pretrained ResNet-18* | | | | | | | | | | | | | |
| SimCLR | 83.56±0.84 | 55.19±1.59 | 33.75±0.25 | 39.01±0.92 | 46.15±0.34 | 60.27±0.66 | 66.85±0.22 | 17.38±0.31 | 27.17±0.84 | 43.12±0.25 | 28.58±0.03 | 45.55±0.27 | 5.73 |
| AugSelf | 84.03±1.31 | 58.65±1.92 | 38.11±0.22 | 41.74±0.92 | 47.80±0.18 | 68.54±0.30 | 69.33±0.36 | 20.23±0.23 | 31.53±0.27 | 45.68±1.19 | 32.51±0.27 | 48.92±0.28 | 4.18 |
| EquiMod | 85.73±0.43 | 60.06±0.68 | 37.43±0.24 | 42.49±1.50 | 48.83±0.10 | 67.07±0.29 | 71.17±0.10 | 19.95±0.69 | 33.03±0.56 | 47.00±0.60 | 32.15±0.10 | 49.54±0.33 | 3.82 |
| CARE | 77.10±0.04 | 51.32±0.01 | **43.52±0.01** | **48.18±0.04** | 46.19±0.02 | 65.84±0.03 | 61.75±0.07 | 21.99±0.00 | 33.77±0.53 | **50.00±0.00** | **35.78±0.00** | 48.68±0.07 | 3.36 |
| STAR-SS | 85.64±0.13 | 61.10±0.10 | 40.17±0.29 | 44.50±0.65 | 50.07±0.27 | 72.59±0.25 | 73.39±0.16 | 21.89±0.29 | 33.90±0.41 | 48.58±0.35 | 34.25±0.28 | 51.46±0.05 | 2.55 |
| STAR-MMoE | **87.45±0.07** | **64.78±0.35** | 41.24±0.16 | 46.82±0.61 | **51.10±0.44** | **73.99±0.25** | **74.76±0.39** | **22.74±0.39** | **35.61±0.21** | 49.75±0.92 | 35.50±0.25 | **53.07±0.21** | **1.36** |

