# OpenReview forum: "Soft Task-Aware Routing of Experts for Equivariant Representation Learning"
_NeurIPS.cc/2025/Conference — NeurIPS 2025 poster_

### Official Review · Reviewer_tSPH · 2025-06-30

**Clarity:** 2
**Significance:** 3
**Originality:** 2
**Rating:** 5
**Confidence:** 4

**Summary:**

This work addresses jointly learning invariant and equivariant representations in self-supervised learning. The authors argue that treating them as fully separate leads to redundancy. Instead of using independent projection heads, they introduce a task-aware routing mechanism that dynamically routes features through shared and task-specific paths. This allows the model to better capture both shared and unique aspects of each objective, improving generalization and reducing representational overlap. Their method shows performance improvements over classification, few-shot learning, and object detection tasks.

**Questions:**

- What would be the point of a separate equivariant learning mechanism? Shouldn’t that be part of the experts’ job? As in, the model should assign an expert to learn the observed shifts in the dataset as its role?

- Since a topic of focus is the separation of heads by adding a shared projector, would you expect forcing disentanglement between representations of separate heads to cause minimal degradation due to the results in figure 3.a?

**Ethical Concerns:**

["NO or VERY MINOR ethics concerns only"]

**Final Justification:**

The proposed revisions to the text the authors have given address everything I was concerned about. The three ablation studies in response to Q1 and Q2 address my concerns about whether there are still some shared concepts left that the explicit mechanisms cannot get rid of. And the justification in the first part of the answer to Q1 appropriately explains why the proposed structure is necessary.

I disagree with other reviewers about the magnitude of performance improvement. This improvement, despite being slightly small in some cases and significant in others, appears to come from the explicit formulation of the proposed method. Considering the extensive ablations that show the efficacy of each part of the proposed method, I’ve raised my score.

**Limitations:**

yes

**Quality:**

3

**Strengths And Weaknesses:**

Strengths:
- The experiments are comprehensive.
- The experiments cover the claims made in the introduction.
- Ablation studies show the effectiveness of the introduced components over previous work.

Weaknesses:
- The introduction section needs to be revised to clearly explain what this paper refers to as invariant and equivariant concepts. I initially assumed that based on figure 1 and L21, the invariance is with respect to a task but there is clear explanation and no buildup or flow in this section. One paragraph is not enough to explain your motivation and intuition before diving to specific examples and details in paragraph 2.

- Following from the previous point, the second paragraph starts with a description of another work that is too technical for the introduction section. Same as before, the concept of invariance and equivariance are being referred to at the same time as invariant and equivariant tasks alongside features. So the introduction needs an overhaul to make it clear.

- About statement in L37, “This not only results in inefficient use of model capacity, but also hinders each head’s ability to specialize in task-specific features.” Say each head learns both shared knowledge and task-specific knowledge. In that case, regardless of redundancy, you still have task-specific features. I’m assuming you’re making a point about heads not being able to only attend to task-specific knowledge and having to share their representational power with shared knowledge but L37 does not explain that clearly and it needs a revision.

---

> ### Author Rebuttal · Authors · 2025-07-31
>
> > W1 & W2. The introduction section needs to be revised to clearly explain what this paper refers to as invariant and equivariant concepts. I initially assumed that based on figure 1 and L21, the invariance is with respect to a task but there is clear explanation and no buildup or flow in this section. One paragraph is not enough to explain your motivation and intuition before diving to specific examples and details in paragraph 2. Following from the previous point, the second paragraph starts with a description of another work that is too technical for the introduction section. Same as before, the concept of invariance and equivariance are being referred to at the same time as invariant and equivariant tasks alongside features. So the introduction needs an overhaul to make it clear.
>
> Regarding the reviewer’s comment on the conceptual clarity of invariant and equivariant learning, we will clarify these concepts in the context of our framework as follows:
>
> - Invariant learning (or task) aims to extract features that remain unchanged under input transformations. It filters out transformation-specific signals while preserving semantic information.
> - Equivariant learning (or task), in contrast, seeks to retain transformation-induced variations in the representation. The goal is to ensure that the representation changes in a structured and predictable manner with respect to the transformation applied.
>
> We clarify that the terms “invariant task” and “equivariant task” refer to **pretext tasks** that induce these respective properties in the learned representations, not to tasks that are themselves invariant or equivariant.  This clarification serves as a conceptual foundation that leads naturally into the second paragraph, where we introduce our modeling strategy and discuss baseline comparisons. We believe this revision significantly improves the clarity and coherence of the introduction.
>
> > W3. About statement in L37, “This not only results in inefficient use of model capacity, but also hinders each head’s ability to specialize in task-specific features.” Say each head learns both shared knowledge and task-specific knowledge. In that case, regardless of redundancy, you still have task-specific features. I’m assuming you’re making a point about heads not being able to only attend to task-specific knowledge and having to share their representational power with shared knowledge but L37 does not explain that clearly and it needs a revision.
>
> We thank the reviewer for the accurate interpretation. We will revise the sentence to better reflect that redundancy reduces the effective capacity each head can dedicate to truly task-specific signals. As the reviewer pointed out, even when using two independent heads, each can still capture both task-specific features and shared knowledge. The core issue, however, lies in the redundancy: both heads must re-encode shared information, which leads to inefficient use of the representational space. Our intention in L37 was to emphasize that this redundancy reduces the effective capacity each head can dedicate to truly task-specific signals. We will revise the sentence to better reflect this point.
>
> > Q1. What would be the point of a separate equivariant learning mechanism? Shouldn’t that be part of the experts’ job? As in, the model should assign an expert to learn the observed shifts in the dataset as its role?
>
> We understand the reviewer’s question as asking whether the explicit equivariant learning mechanism is necessary, given that the MoE gating could in principle assign experts to learn augmentation-induced shifts as part of their specialization. Based on this understanding, we provide the following response.
>
> 1. Gating Alone Does Not Induce Equivariance
>
> We agree that the mixture-of-experts (MoE) mechanism encourages each expert to focus on different portions of the data distribution. However, gating only selects which expert to use; it does not constrain the functional role of each expert. In particular, gating does not ensure that the selected expert encodes augmentation-specific transformations in a consistent and structured manner without equivariant objective.
>
> 2. Why An Explicit Equivariant Objective Is Necessary
>
> The invariant loss encourages the model to remove augmentation-induced variations to enforce semantic consistency. This removes the signals, such as color and pose, needed to learn augmentation-aware representations. In contrast, the equivariant objective provides an **explicit inductive bias** that encourages experts to preserve these signals in the embedding space by modeling transformation-consistent feature shifts.
>
> 3. Gating Without Equivariant Loss
>
> To isolate the effect of the equivariant objective, we performed an ablation where we removed the equivariant loss while keeping the MoE architecture with one gating network. The result shows that gating alone is insufficient for learning robust, generalizable features:
>
> | Method | In-domain | Out-domain |
> | :-: | :-: | :-: |
> |  MMoE (Ours) | 86.74 | 53.07 |
> | Invariance-only MoE | 85.93 | 49.27 |
>
> Despite identical architecture (except for gating network) and capacity, the performance drops significantly without the equivariant loss. This suggests that without an equivariant loss, the experts do not specialize in modeling augmentation-specific information, even when gating is present.
>
> We hope this addresses the reviewer’s question. If we misunderstood the question, we kindly ask the reviewer to rephrase or elaborate so we may provide a more precise response.
>
> > Q2. Since a topic of focus is the separation of heads by adding a shared projector, would you expect forcing disentanglement between representations of separate heads to cause minimal degradation due to the results in figure 3.a?
>
> The observed expert-wise separation in **Figure 3(a)** suggests that additional constraints enforcing disentanglement across heads would have limited effect, as our method already induces clear task-specific specialization via the MMoE gating mechanism.
>
> To verify this, we conducted two complementary experiments:
>
>
> - Orthogonality regularization: We explicitly enforced orthogonality across expert outputs and observed only a small drop in out-domain classification accuracy from **53.07** to **52.75**, confirming that he additional orthogonality constraint introduces only minor adjustments to the parameters.
> - Hard assignment: We replaced soft gating with hard expert selection (1 shared + 5 invariant + 2 equivariant), matching the effective expert allocation observed in Figure 3(a). The resulting model showed only a minor performance drop from **52.37 (MMoE with 8 experts)** to **52.27**, indicating that the learned gating already approximates hard selection.
>
> These results support the conclusion that our MMoE structure effectively promotes disentangled, task-specific representation learning, even without explicit constraints.
>
> We hope our response aligns with the reviewer’s intent, and we would welcome any clarification if we have misunderstood the question.

---

> > ### Comment · Reviewer_tSPH · 2025-08-04
> >
> > Thank you for the proposed revisions and the new experiments. All my concerns have been addressed. I'll raise my score to 5.

---

> > > ### Author Response · Authors · 2025-08-05
> > >
> > > We are glad to hear that all your concerns have been addressed. We will revise our manuscript based on your suggestions to further improve its clarity in the final version.

---

### Official Review · Reviewer_5sNH · 2025-06-30

**Clarity:** 4
**Significance:** 3
**Originality:** 2
**Rating:** 5
**Confidence:** 3

**Summary:**

The paper tackles the interesting problem of representational redundancy in self-supervised learning methods using both invariant and equivariant objective functions. The authors point out that predicting semantic category and predicting augmentation may require similar features. Training projection heads for each task independently may therefore cause similar features to be encoded multiple times in the representation, reducing overall model capacity. The authors propose a mixture-of-experts based gating mechanism in which several shared projection heads are trained simultaneously. Objective-based gating networks produce softmax distributions over the experts to appropriately weight the most fitting projections for each task. This, in addition to an equivariant learning mechanism using a residual connection, yields representations that produce superior empirical results to comparable equivariant learning methods. Additional analysis of the networks trained using their mechanism demonstrate some evidence that redundancy among projection heads is reduced can lead to poorer performance.

**Questions:**

- [W2] Are all encoders trained for 200 epochs (Section 5, setup)? Figure 5 potentially addresses this, but what about the other experiments in the paper?
- [W4] Is the argument about representational redundancy concerning the encoder f(x) itself, or the outputs of the projection heads? Couldn’t it be the case that both the invariant and equivariant projection heads could use the same feature in the representation f(x) without that feature being encoded multiple times in the representation? Is the issue then concerning the capacity of the projection heads themselves?
- [W3] Is there any justification for why the new method can hurt in-domain performance?

**Ethical Concerns:**

["NO or VERY MINOR ethics concerns only"]

**Final Justification:**

Authors addressed each of the four listed weaknesses, providing additional experiments.

**Limitations:**

Pending the weaknesses mentioned previously, the authors adequately discuss the limitations of their work.

**Paper Formatting Concerns:**

No concerns.

**Quality:**

3

**Strengths And Weaknesses:**

Strengths
- The paper is well written, easy to follow, and has easy to interpret figures.
- A large number of experiments are run, combining various tasks, datasets, evaluation types (in-domain, out-of-domain), and baselines.
- The additional analyses presented in section 5.2 help the reader understand how different kinds of experts naturally emerge using the new mechanism.
- The practicality of the method is considered, with Figure 5 demonstrating that the new method can produce stronger results in a shorter time.

Weaknesses
- [W1] The empirical gains from using the method are often quite small.
- [W2] Not all comparisons may be fair. Are all encoders trained for 200 epochs (Section 5, setup)? If so, this would mean that the author’s method gets a significant computational advantage, since the method “incur(s) a higher cost per epoch”.
- [W3] It is not clear why the method could actually hurt in-domain performance under some circumstances.
- [W4] I’m not entirely convinced that redundancy/canonical correlation amongst the projection from the experts implies redundancy in the un-projected representation. See questions.

---

> ### Author Rebuttal · Authors · 2025-07-31
>
> >  [W1] The empirical gains from using the method are often quite small.
>
> We acknowledge the reviewer’s concern. Our method shows clear improvements across multiple benchmarks, and the performance gains are comparable to those of prior methods. For example, on STL10, we achieve a **7.1%** relative gain over the previous best (EquiMod) while attaining the highest overall accuracy. On ImageNet100, we observe **1.3%** gain over EquiMod.
>
> Similar trends are observed in the few-shot classification and object detection tasks (see **Table 3-4**), where our method consistently outperforms the strongest baseline. These results collectively demonstrate the effectiveness of our approach across diverse downstream tasks.
>
> > [W2] Are all encoders trained for 200 epochs (Section 5, setup)? Figure 5 potentially addresses this, but what about the other experiments in the paper?
>
> Unless otherwise stated, all encoders are trained for 200 epochs. We chose this schedule because 200 epochs allows the baselines to reach their best performance and yields reasonable performance for our method as well.
>
> Importantly, as shown in **Figure 5**, **our methods already outperforms the strongest baseline even when trained for significantly fewer epochs, and its performance continues to improve with longer training**.
>
> A similar trend is observed in few-shot classification and object detection. In particular, under the few-shot classification setting, our method, trained with only 100 epochs, still outperforms the prior best method (AugSelf) on STL10-pretrained ResNet-18. Specifically, while AugSelf achieves an average accuracy of 53.54% (Table 2), our method reaches **55.27%**, yielding a relative improvement of **3.23%** despite fewer training iterations. This result **highlights the sample efficiency and strong transferability of our approach**.
>
> > [W3] Is there any justification for why the new method can hurt in-domain performance?
>
> We attribute the slight drop in STL10 in-domain accuracy to the difference in dataset complexity and the effect of increased model capacity.
>
> Compared to EquiMod, our method adopts a **deeper predictor** to better model complex augmentation-induced shifts, which enhances out-domain generalization. However, in low-diversity datasets like STL10, where semantic variation is relatively limited, the added capacity may introduce excessive degrees of freedom, leading to unnecessary feature dispersion and marginal degradation in in-domain accuracy.
>
> To support this explanation, **Table 5** presents an ablation using a shallower predictor (depth = 1). In this setting, the in-domain performance improves to **87.41%**, confirming that deeper predictor can be a factor in performance drop under simpler data conditions.
>
> Importantly, in the context of SSL, the quality of learned representations is typically assessed by their transferability rather than in-domain performance. From this perspective, our architectural choice prioritizes generalization, as evidenced by consistently strong out-domain classification results.
>
> > [W4] Is the argument about representational redundancy concerning the encoder f(x) itself, or the outputs of the projection heads? Couldn’t it be the case that both the invariant and equivariant projection heads could use the same feature in the representation f(x) without that feature being encoded multiple times in the representation? Is the issue then concerning the capacity of the projection heads themselves?
> We clarify that the redundancy we discuss arises at the projection head output level. However, this head-level redundancy negatively impacts the encoder.
> This may be harmless at inference, but during training, independent heads must each disentangle both shard and task-specific factors from entangled backbone features. This is inherently difficult and often results in suboptimal gradients being propagated back into the encoder, degrading representation quality.
>
> We clarify that the redundancy we discuss arises at the projection head output level. However, this head-level redundancy negatively impacts the encoder.
>
> This may be harmless at inference, but during training, independent heads must each disentangle both shard and task-specific factors from entangled backbone features. This is inherently difficult and often results in suboptimal gradients being propagated back into the encoder, degrading representation quality.
>
> In contrast, explicitly separating shared and task-specific pathways (e.g., via a shared projector or MMoE gating) allows heads to specialize more easily. They converge faster and provide more reliable gradients. We confirmed this empirically: the Frobenius norm of differences in projector weights every 10 epochs decreased more quickly with our method than with EquiMod, indicating more stable optimization.
>
> | Method        | 20   | 30   | 40   | 50   | 60   | 70   | 80   | 90   | 100  | 110  | 120  | 130  | 140  | 150  | 160  | 170  | 180  | 190  | 200  |
> | :-: | :-:  | :-:  | :-:  | :-:  | :-:  | :-:  | :-:  | :-:  | :-:  | :-:  | :-:  | :-:  | :-:  | :-:  | :-:  | :-:  | :-:  | :-:  | :-:  |
> | EquiMod       | 7.16 | 5.05 | 4.19 | 3.74 | 3.47 | 3.27 | 3.07 | 2.88 | 2.66 | 2.43 | 2.18 | 1.91 | 1.63 | 1.33 | 1.02 | 0.72 | 0.44 | 0.19 | 0.04 |
> | Single Shared | 6.58 | 4.61 | 3.86 | 3.54 | 3.27 | 3.13 | 2.98 | 2.81 | 2.58 | 2.37 | 2.10 | 1.85 | 1.59 | 1.30 | 1.00 | 0.71 | 0.42 | 0.19 | 0.04 |
> | MMoE          | 5.28 | 3.13 | 2.21 | 1.88 | 1.68 | 1.55 | 1.45 | 1.35 | 1.25 | 1.15 | 1.03 | 0.91 | 0.78 | 0.64 | 0.50 | 0.36 | 0.22 | 0.10 | 0.02 |
>
> Finally, this issue cannot be explained by projection head capacity alone. Even when we increased the number of projectors from 2 to 8 while reducing the hidden dimension (512 →180) to keep the parameter count approximately the same, applying MMoE Projection still improved the performance from **49.54** to **50.34**. This demonstrates that the performance gain comes from the inductive bias introduced by MMoE, not simply from increasing capacity.

---

> > ### Comment · Reviewer_5sNH · 2025-08-01
> >
> > Thank you for addressing my concerns. I have no further questions regarding [W3, W4]. I further respond to your comments on [W1, W2] below.
> >
> > [W1, W2]
> >
> > The diversity of tasks considered is definitely a strength of the paper. However, for a method that achieves marginally superior average performance on a number of tasks, while also degrading performance on some tasks (the STL-10 trained model achieves the 2nd best performance on 4/11 tasks in Table 1 and performs worse in-distribution), I would need to see a more thorough investigation to improve my score. There are two ways in particular I am thinking of:
> >
> > 1. Control for amount of computation used across all experiments. It seems like the method can still present marginal gains when compute is controlled for in some settings, but confirming that this is true for each of the circumstances in the paper would present a convincing argument for this being a practical self-supervised learning method.
> > 2. Report the hyperparameter tuning procedure used for each method. I would suspect that some hyperparameter tuning was done for the method presented in the paper based on the fact that different numbers of experts are used depending on the pretraining dataset used, and section A.4 in the appendix. Ensuring that all the baselines received a similar amount of hyperparameter tuning would also be very convincing.

---

> > > ### Author Response · Authors · 2025-08-05
> > >
> > > Thank you for further engaging in the discussion. Regarding the (slightly) lower in-distribution performance of our method compared to some prior works, we believe this is not a significant drawback considering the performance improvement in out-domain generalization, indicating that *prior works have a tendency to overfit to the training distribution and a limited ability to generalize to unseen domains,* as we discussed in L284.
> > >
> > > > Q1. Control for amount of computation used across all experiments. It seems like the method can still present marginal gains when compute is controlled for in some settings, but confirming that this is true for each of the circumstances in the paper would present a convincing argument for this being a practical self-supervised learning method.
> > >
> > > We agree that demonstrating consistent performance gains under fair conditions is important. To summarize how we address this so far:
> > >
> > > 1. **Figure 5** addresses this concern with varying **training time**
> > >
> > > 2. Our response to [W2] provides a comparison where baseline methods are given more training time.
> > >
> > > 3. Our response to [W4] reports the result when controlling the number of parameters of projectors, ensuring the number of parameters is approximately matched across compared methods.
> > >
> > > However, the notion of *fair condition* might differ depending on context; for example, we can try to *match* training time or model size (which is tricky for some baselines), or we can *ensure the best performance* via validation by considering them as tunable hyperparameters (e.g., early stopping). Notably, in terms of the best out-domain performance (which we mainly focus on throughout evaluation), we actually provide an *unfair advantage* to baseline methods, in that their performance peaks around 200 epochs and subsequently decreases, while our method continues to improve with further training.
> > >
> > > Taking account of your concern, in the revision, we will provide more controlled experiments by ensuring fair comparisons in different aspects. Due to the limitations in our computational resources, we cannot present them all now; in any case, we are running experiments and plan to incorporate them in the revision. We also welcome your opinion on *what should be fair to make it more convincing*, as we believe it should be a valuable guidance to strengthen our work.
> > >
> > > Below we provide some additional experimental results:
> > >
> > > **Matching # params:** We conducted an additional experiment on ImageNet100. Specifically, we used 8 experts with a reduced hidden dimension (2048 → 724), resulting in a total parameter count comparable to that of EquiMod [1]’s projectors.
> > >
> > > | Method | # params for projectors | In-domain | Out-domain | Few-shot |
> > > | :-: | :-: | :-: | :-: | :-: |
> > > | EquiMod | 17.31M | 83.23 | 69.10 | 58.97 |
> > > | **MMoE (Ours, dim=724)** | **16.81M** | **83.44** | **69.48** | **60.77** |
> > >
> > > The columns correspond to in-domain accuracy, out-domain average accuracy, and few-shot average accuracy, respectively. (We skip object detection as it takes a long time.) These results suggest that our improvements hold even under compute-controlled conditions beyond the STL10 setting.
> > >
> > > **Varying training time:** We conducted an additional experiment on ImageNet100 where EquiMod [1] trained for 200 and 250 epochs, while our method trained for 200 epochs. Although EquiMod [1] requires less wall-clock time per epoch (4m 51s vs. 5m 20s), training it for 250 epochs results in a greater overall compute budget. Nonetheless, our method achieves superior performance under compute-controlled conditions.
> > >
> > > | Method | Training time | In-domain | Out-domain | Few-shot |
> > > | :-: | :-: | :-: | :-: | :-: |
> > > | EquiMod (200 epochs) | 16hr 10m | 83.23 | 69.10 | 58.97 |
> > > | EquiMod (250 epochs) | 20hr 12m | 83.12 | 69.27 | 59.95 |
> > > | **MMoE (Ours, 200 epochs)** | **17hr 47m** | **83.43** | **70.01** | **63.50** |

---

> > > > ### Comment · Reviewer_5sNH · 2025-08-05
> > > >
> > > > Thank you for the response. I appreciate the details regarding the hyperparameter tuning, as it helps to ensure that this comparison is fair. Please be sure to include these notes in future versions of the manuscript.
> > > >
> > > > Regarding computational constraints, I think the most fair way to compare methods in this case is to control for total compute budget. I think wall clock time or flops are both good means of measuring this, as parameter counts may be misleading if a more complicated loss function is computationally taxing. While the baselines seem to achieve their greatest performance around 200 epochs in Figure 5, in general one would expect an increased compute budget to potentially lead to better generalization.
> > > >
> > > > It is understandable that recreating each of the experiments detailed in the paper in a compute-constrained manner is not possible under the time constraints of the discussion period. This should be added to the final manuscript to convincingly demonstrate MMoE's practicality. That being said, the supplementary experiments controlling for clock time as well as parameter counts have been convincing, and I will increase my score.

---

> > > > > ### Author Response · Authors · 2025-08-06
> > > > >
> > > > > Thank you for your helpful suggestions, especially regarding fair comparisons in terms of compute budget and hyperparameter tuning. We will revise our manuscript accordingly and include additional experiments to address these points. We appreciate your positive feedback.

---

### Official Review · Reviewer_inzh · 2025-07-01

**Clarity:** 2
**Significance:** 2
**Originality:** 2
**Rating:** 4
**Confidence:** 3

**Summary:**

This works introduces the use of Mixture of Experts module in place of the standard MLP projector in a SSL framework. The authors present this framework as a mechanism to support equivariant representation learning as the MoE allows for specialisation to invariant and equivariant tasks either independently or as a shared space of the two. They introduce a series of constructive losses to enforce both representation properties, with the equivariant component enforced with a predictor network. The authors report strong invariant performance on a variety of benchmark tasks, and demonstrate robustness in and out of distribution. The ablation studies perform an investigation into the behaviour of the MoE and its impact on equivariant loss.

**Questions:**

- How does the k-nn retrieval perform on the output of each expert? Do you get specialised transformation encodings?
- $\psi$ from figure 2 doesn’t seem to be defined in the text, I assume this is some network to encoder the transformation parameters.
- Is $\phi_{T}$ a mlp predictor or a different type of network? This is not defined.
- Why is there a drop in performance as a factor of training time with SimCLR (Figure 5), from experience with SimCLR this behaviour is unusual and instead the convergence behaviour of equimod is expected. While you state overfitting, the mean accuracy is low. Can you also confirm which dataset this is performed on?

**Ethical Concerns:**

["NO or VERY MINOR ethics concerns only"]

**Final Justification:**

I have increased my score as the authors have presented some additional results that address my concerns. I however, cannot comfortably give a higher score due to the concerns raised by other reviewers including limited improvement over baseline and large number of changes needed on the full manuscript.

**Limitations:**

Limitations are addressed, but could be expanded to include task specifics and broader implications.

**Paper Formatting Concerns:**

No formatting concerns.

**Quality:**

3

**Strengths And Weaknesses:**

**Strengths**:
+ The investigation of equivariant SSL is timely and important to the computer vision community.
+ The method presented is sensible, seemingly correct, and simple. It is presented as general purpose method that can be adapted to a variety of data domains, architectures and SSL transformations.
+ The invariant representation quality is well explored with a wide variety of experimentation, tests for robustness and generalisation.
+ The investigation into redundancy is a nice addition, and demonstrates well that the experts are able to specialise and improve performance without wasted resource.

**Weaknesses**:
- The main argument made at the outset of the paper is that using separate projection heads limits the expressivity of the network and limits shared information. However, your findings in Figure 3 (a) somewhat contradict your conjecture. In fact it can be seen that from the expert gating that experts are largely specialised to either invariant or equivariant, with only one being shared, but not balanced. Therefore, one could argue that the improved results are due to the increased capacity for the network to learn specialised equivariant and invariant features given MoE. It would be nice to prove that simply having a projector network for Transformation type in T does not reach the same performance, given that T is accessible during training.
- The evaluation does not analyse or report any quantitative measures for equivariant representations. Therefore, there is no way to know if the objective function is actually learning equivariant representations. Some further analysis of the learnt representation space is needed before the claim of equivariance can be substantiated. For example, all experimentation evaluates the invariance of the representations, yet does not evaluate explicitly the transformation information in the representations.
- The novelty of the approach is arguably limited, this method is essentially akin to EquiMod, or SEN but with a MoE as a projector. While I am keen for these findings to be distributed to the wider community, the novelty is narrow.
- While it is nice to show faster convergence with wall time, it would be beneficial to see the memory consumption, iterations per second, FLOPs as a more quantitative evaluation of computer resources.
- There are some omitted details in the manuscript that have been outlined in the questions, these is a minor concern that will improve the reproducibility of the work if included.

---

> ### Author Rebuttal · Authors · 2025-07-31
>
> > W1. The main argument is that separate projection heads limit expressivity of the network and limits shared, but Figure 3(a) seems to contradict this. One could argue the gains come from increased capacity to learn specialized features. It would be nice to show that a projector per transformation type T would not match the same performance.
>
> The imbalance observed in **Figure 3(a)**, where only one expert is shared across tasks, does not contradict our conjecture. Our argument is not that experts must be balanced between task, but that having two independent projectors leads to redundant learning of shared information. In MMoE Projection, the observed specialization naturally emerges during optimization as the model allocates experts according to the invariant and equivariant objectives while avoiding repeated extraction of shared factors.
>
> To address the possibility that the improved results simply stem from increased capacity, we conducted an ablation study using multi-projector baseline with six dedicated projection heads: one for invariance and the others for five transformation types. For fairness, the MMoE variant was also trained with six experts to match model capacity.
>
> | Method | Acc |
> | :-: | :-: |
> | EquiMod | 49.54 |
> | Multi-Projector  | 49.61 |
> | MMoE (6 Experts) | 52.35 |
>
> Despite identical capacity, MMoE Projection shows a relative improvement of **5.5%** out-domain classification performance than the transformation-specific multi-projector baseline (52.35 vs. 49.61). This confirms that the gain arises not merely from increased capacity, but from the structured inductive bias and task-aware routing introduced by MMoE.
>
> > W2. The evaluation does not analyse or report any quantitative measures for equivariant representations. Some further analysis of the learnt representation space is needed before the claim of equivariance can be substantiated.
>
> Our paper provides clear evidence that the learned features preserve augmentation-aware (i.e., equivariant) information, following the evaluation paradigm established in prior work such as AugSelf [3], where color consistency is used as a key qualitative indicator of equivariance. Based on this criterion, we provide two complementary pieces of evidence:
>
> - Figure A.2: Retrieves neighbors with similar color attributes in k-NN retrieval.
> - Table 1: Achieves the highest accuracy on the color-sensitive Flowers102 dataset.
>
> These results indicate that our method effectively captures transformation-sensitive features.
>
> To further strengthen this claim, we add explicit quantitative evaluations of transformation-aware information in the learned representations:
>
> 1. Transformation Parameter Regression
> We design a regression task using a frozen backbone and a single linear layer to predict augmentation parameters given the original and augmented images. We evaluate performance using Mean Squared Error (MSE).
>
> - For Crop, we predict scale and aspect ratio.
> - For Color, we predict brightness, contrast, saturation, and hue.
> - When both are applied, we predict all parameters jointly.
>
> | Method   | Crop ↓ | Color ↓ | All ↓  |
> | :-: | :-: | :-: | :-: |
> | SimCLR   | 0.016  | 0.026   | 0.025  |
> | AugSelf  | 0.006  | 0.017   | 0.016  |
> | EquiMod  | **0.005**  | 0.020   | 0.016  |
> | **MMoE (Ours)** | **0.005**  | **0.013**   | **0.013**  |
>
> MMoE achieves the lowest MSE across all transformation types, indicating that its representations more accurately encode transformation-aware information.
>
> 2. Transformation Type Classification
> We also perform a binary classification task to distinguish whether the applied transformation is Crop or Color, using the same frozen representation and a linear head.
>
> | Method   | Acc (%) ↑ |
> | :-: | :-: |
> | SimCLR   | 82.34      |
> | AugSelf  | 91.74      |
> | EquiMod  | 92.79      |
> | **MMoE (Ours)** | **95.76**  |
>
> MMoE attains the highest classification accuracy (**95.76%**), suggesting that it preserves clear discriminative signals between different transformation types in the learned representation.
>
> These results collectively provide both **quantitative** and **qualitative** support for the claim that our method captures **transformation-aware** features in the learned representations.
>
> > W3. The novelty of the approach is arguably limited, this method is essentially akin to EquiMod, or SEN but with a MoE as a projector. While I am keen for these findings to be distributed to the wider community, the novelty is narrow.
>
> Our method proposes projection module designs that addresses the limitations of prior work such as EquiMod [1] and SEN [2] by explicitly handling redundancy and improving specialization:
>
> - Diagnosing the redundancy problem between two independent heads in Equivariant SSL.
> - Structuring the projection space to separate shared and task-specific components.
> - Leveraging multi-gate routing aligned with invariance and equivariance objectives.
>
> These designs lead to better expert specialization and improved downstream performance, offering both conceptual and practical advances over prior work.
>
> > W4. While it is nice to show faster convergence with wall time, it would be beneficial to see the memory consumption, iterations per second, FLOPs as a more quantitative evaluation of computer resources.
>
> Thank you for the suggestion. We report a detailed comparison of memory usage, FLOPs, and iteration speed in the table below. While MMoE Projection has marginally higher memory consumption and FLOPs than EquiMod, the values remain largely comparable.
>
> | Method      | Iter/s ↓  | Memory (MB) ↓  | GFLOPs ↓  | Acc  ↑  |
> | :-: | :-: | :-: | :-: | :-: |
> | SimCLR      | 17.03 | 3272       | 0.67         | 45.55 |
> | AugSelf     | 16.78     | 3303           | 0.67             | 48.92 |
> | EquiMod     | 12.62     | 4343           | 1.01             | 49.54 |
> | MMoE (Ours) | 9.69      | 4484           | 1.03             | 53.07 |
>
> Importantly, as shown in **Figure 5**, MMoE Projection achieves strong performance even with fewer training iterations, demonstrating that the model is not only effective but also sample-efficient. This confirms that its improved performance stems not from increased computational cost, but from architectural advantages.
>
> > W5. There are some omitted details in the manuscript that have been outlined in the questions, these is a minor concern that will improve the reproducibility of the work if included. (Q2 & Q3 included)
>
> We are committed to ensuring reproducibility, and to that end, we have included not only the definitions of $\psi$ and $\phi_T$, but also all relevant hyperparameter settings and configuration details in **Appendix B**. To make the manuscript more self-contained, we will move key definitions such as those of $\psi$ and $\phi_T$ into the main text in the revised version.
>
> Specifically, as defined in **Appendix B.2**:
>
> - $\psi$ is a **single linear layer** that takes the augmentation parameter vector as input and expands it to match the output dimensionality of the expert representations. This design follows EquiMod and emphasizes the role of augmentation parameters in guiding the transformation.
>
> - $\phi_T$ is an **MLP predictor**.
>
> > Q1. How does the k-nn retrieval perform on the output of each expert? Do you get specialised transformation encodings?
>
> Our analysis suggests that the experts are not specialized by transformation type, but rather align more closely with objective-level distinctions (i.e., shared vs. invariant vs. equivariant), as indicated by their k-NN retrieval results (**Figure A.3**).
>
> To analyze this, we conducted k-NN retrievals within each expert’s embedding space on the STL10 test set. The results show that:
>
> - Expert 1 (shared) retrieves **semantically consistent images**, reflecting general-purpose representation.
> - Experts 2–6 retrieve samples **invariant to changes induced by transformation**, aligning with the invariant objective.
> - Experts 7–8 retrieve samples with **similar color tone**, suggesting sensitivity to color-based transformations.
>
> This suggests that the experts specialize based on the nature of the objective, rather than being explicitly aligned with specific transformation types.
>
> > Q4. Why is there a drop in performance as a factor of training time with SimCLR (Figure 5), from experience with SimCLR this behaviour is unusual and instead the convergence behaviour of equimod is expected. While you state overfitting, the mean accuracy is low. Can you also confirm which dataset this is performed on?
>
> The performance drop with longer training in **Figure 5** is observed because we evaluate **out-domain generalization**, not in-domain performance. As shown in **Figure A.1(a)** in Appendix, the in-domain accuracy consistently improves with longer training, while out-domain accuracy degrades. The out-domain results in **Figure 5** are evaluated on a **collection of 11 out-domain datasets** (details in **Appendix B.1**). This indicates that the drop is not a convergence problem but instead reflects a loss of generalization, likely stemming from the lack of explicit inductive bias in SimCLR for modeling augmentation-aware structures.
>
> We will consider including the corresponding in-domain performance curve (currently in **Figure A.1(a)**) in the main text in the revised version for clarity.
>
> ### Reference
> [1] Devillers and Lefort. EquiMod: An Equivariance Module to Improve Self-Supervised Learning. In *ICLR*. 2023.
>
> [2] Park et al. Learning Symmetric Embeddings for Equivariant World Models. In *ICML*. 2022.
>
> [3] Lee et al. Improving Transferability of Representations via Augmentation-Aware Self-Supervision. In *NeurIPS*. 2021.

---

> ### Comment · Reviewer_inzh · 2025-08-04
> **Response to rebuttal**
>
> Thank you to the authors for their clarifications on these points, and apologies for the misunderstanding of Q1 and Q4.
>
> My only concern still lies with the evaluation of equivariance, while results are presented on colour specific tasks and that there is some retrieval based evaluations, this is not necessarily cause for determining equivariance. Instead it can be argued that the work is augmentation aware, not necessarily equivariant without providing theoretical proof, or at the very least some indication that the latent space can be predictably transformed inversely to the input transformations.
>
> I understand this may seem pedantic but the definition of equivariance here is not substantiated.
>
> Given that my other concerns have been addressed, I will increase my score.

---

> > ### Author Response · Authors · 2025-08-06
> >
> > Thank you for further engaging in the discussion.
> >
> > We acknowledge your concern that our previous results, such as k-NN retrieval, transformation parameter regression, and transformation type classification, may primarily reflect augmentation-awareness rather than quantifying equivariance.
> >
> > We recently found a concurrent work [1], which might be useful to address this concern. We adopt the **representation Equivariance (R-equivariance)** evaluation protocol, which learns a predictor on top of the frozen backbone for a downstream task of *predicting the resulting representation when applying the transformation to inputs* to measure whether learned representations **predictably reflect** input-space transformations. This should be a good way to measure whether the latent space encodes equivariance in a predictable way.
> >
> > Concretely, we freeze the backbone and train a predictor takes **the original representation and the transformation parameters** as an input, and predicts **the representation of the transformed image**. The cosine similarity between the predicted and actual representations is used as the equivariance score.
> >
> > Additionally, we extend the original protocol–which considers a single parameter at a time (e.g., brightness or contrast)–to jointly handle multiple transformation parameters, providing a more comprehensive and challenging evaluation.
> >
> > As shown below, our method achieves *near-perfect equivariance*, outperforming all prior methods:
> >
> > | Method   | R-equivariance ↑  |
> > | :-: | :-: |
> > | SimCLR   | 0.74  |
> > | AugSelf  | 0.92  |
> > | EquiMod  | 0.91  |
> > | **MMoE (Ours)** | **0.98**  |
> >
> > We will incorporate this in the revised manuscript, as it reflects the notion of equivariance more directly..
> >
> > ### Reference
> > [1] Plachouras et al. Towards a Unified Representation Evaluation Framework Beyond Downstream Tasks. In *IJCNN*. 2025.

---

### Official Review · Reviewer_TqVZ · 2025-07-03

**Clarity:** 3
**Significance:** 2
**Originality:** 3
**Rating:** 4
**Confidence:** 3

**Summary:**

This paper introduces a new Invariant-Equivariant SSL paradigm that combines the invariant and equivariant losses through a mixture of experts’ approach. This allows the model to learn both components, but removes the redundant information across the two projector heads. Some of the learned projector heads become specialized for one task while others are shared across the tasks. The authors present experimental results on STL10 and Imagenet100 pretrained models, testing a wide range of out-of-domain classification tasks, and find that their method improves out of domain classification.

**Questions:**

* I found Figure 1 confusing. Naively, I would think the L/R flip of the objects would also be preserved as an augmentation, but this stands out as something that the invariance-only model is doing better at preserving than the equivariant model.  Could this be explained, or could the figure be made clearer? For instance, what would hypothetical images be that visually contain the augmentation-aware information but do NOT contain semantic structure?
* The paper should cite and discuss CE-SSL from last years’ NeurIPS, which also combines equivariant and invariant losses (this is the paper that also talks about ImageNet1000 vs ImageNet100 training for equivariant model evaluation) https://proceedings.neurips.cc/paper_files/paper/2024/file/ae28c7bc9414ffd8ffd2b3d454e6ef3e-Paper-Conference.pdf
* The authors talk about “representational redundancy” and how it is reduced by including a shared projector in the invariant and equivariant projectors, however, to my knowledge, these projectors are typically discarded anyway and the “representation” that is typically used is the backbone (which is shared across the different projectors). How does this learned representation (the backbone representation) relate to the assumption of independence of the projectors? Naively, I am not sure why having completely independent projectors should encourage one to learn a better representation, and as the text is currently written, the more general motivation for benefits of this change are not clear.

**Ethical Concerns:**

["NO or VERY MINOR ethics concerns only"]

**Final Justification:**

During the rebuttal the authors detailed changes that would be made for improving the clarity of the submission and placing it in context of other work. The authors also detailed additional analyses to look at the equivariant and invariant heads of the models, and measured how the learning dynamics changed with the various variants of the loss.

Overall, I think this is a solid submission and appropriate for the conference, but the number of changes required has me a little hesitant to give it a higher score without seeing an updated draft.

**Limitations:**

yes

**Quality:**

3

**Strengths And Weaknesses:**

**Strengths**
* Incorporating MMoE into the projection model to select the “task” by using a gating network seems quite interesting as a way to mitigate the dependence on the lambda term for the two losses.
* The paper presents results with other invariant learning objectives (Table 6) to show that the results generalize in other SSL frameworks.

**Weaknesses**
* The paper motivation states that the invariance and equivariant components are encoding “redundant” information, however this statement is not quantified. It would be nice to have seen this quantified in some way for the comparison models, and then show that it is improved for the presented models. The k-NN retrieval is qualitative and not quantitative.
* The analysis of the “expert specialization” is performed only on the STL10 network, and the results seem somewhat qualitative and difficult to interpret.
* Although the results for out-of-domain classification seem very reasonable and consistent, suggesting the model is doing something interesting, the margin of differences across models is very small for these OOD experiments (and has been reported to disappear for the equivariant models when trained on ImageNet1000, see below).

---

> ### Author Rebuttal · Authors · 2025-07-31
>
> > W1. The paper motivation states that the invariance and equivariant components are encoding “redundant” information, however this statement is not quantified. It would be nice to have seen this quantified in some way for the comparison models, and then show that it is improved for the presented models. The k-NN retrieval is qualitative and not quantitative.
>
> We already quantify representational redundancy using the **canonical correlation** between the heads, as shown  in **Figure 4**. This metric captures the linear dependency between representations produced by different heads and provides a direct measure of redundancy across projection spaces. The connection between head-level redundancy and representation is further discussed in our response to **Q3**.
>
> As reported in **Figure 4(b)**, EquiMod [1], which uses two independent projectors for invariant and equivariant learning, exhibits high Canonical Correlation (0.5830), indicating substantial overlap in learned embeddings. In contrast, our methods, including Single Shared Projection (0.5383) and MMoE Projection with 16 experts (0.3948), show significantly lower Canonical Correlation. This corresponds to a **7.6%** and **32.3%** relative reduction, respectively, reflecting reduced redundancy and enhanced specialization across heads.
>
> > W2. The analysis of the “expert specialization” is performed only on the STL10 network, and the results seem somewhat qualitative and difficult to interpret.
>
> We further confirmed expert specialization patterns in our analysis on ImageNet100, which also show clear and interpretable expert allocation. We will include the ImageNet100 results in the revised version.
>
> **Figure 3** provides both **quantitative** and **qualitative** evidence of expert specialization.
>
> Specifically, **Figure 3(a)** visualizes the gating weight distribution across experts, showing that during training, the model naturally assigns distinct roles: 1 shared expert, 5 invariant experts, and 2 equivariant experts among the 8 experts. The Min/Max ratio shows how evenly each expert is used across invariant and equivariant learning tasks. For instance, Expert 1, identified as the shared expert, has a Min/Max ratio of **0.34**, indicating that it is assigned to both tasks relatively evenly. This balanced assignment encourages the expert to capture shared information beneficial to both objectives.
>
> **Figure 3(b)** and **Figure A.3** in Appendix visualize the k-NN retrieval results based on each expert’s embedding. Invariant experts retrieve samples with similar shapes, whereas equivariant experts tend to focus on color similarity. The shared expert, on the other hand, retrieves semantically consistent samples within the same class. These results indicate that different experts attend to distinct transformation sensitivity and semantic attributes, providing clear evidence of expert specialization.
>
> > W3. Although the results for out-of-domain classification seem very reasonable and consistent, suggesting the model is doing something interesting, the margin of differences across models is very small for these OOD experiments (and has been reported to disappear for the equivariant models when trained on ImageNet1000, see below).
>
> As commonly reported in the SSL and Equivariant SSL literature, performance gains on OOD tasks are often  marginal.
>
> For instance, SimSiam [2] reports similarly small margins (<1% absolute) over the prior best (e.g., MoCo v2) on VOC/COCO transfer learning tasks, yet these consistent gains are considered indicative of improved generalization.
>
> > Q2. The paper should cite and discuss CE-SSL from last years’ NeurIPS, which also combines equivariant and invariant losses (this is the paper that also talks about ImageNet1000 vs ImageNet100 training for equivariant model evaluation)
>
> We thank the reviewer for highlighting CE-SSL [3]. CE-SSL introduces a contrastive equivariant SSL learning framework that explicitly adds equivariance-related objective to complement standard invariant SSL losses. Unlike our method, which proposes a new projection module design employing MMoE to reduce redundancy between projectors, CE-SSL [3] addresses the problem at the loss-design level.
>
> We will revise the manuscript to cite and discuss this work. Regarding the findings on ImageNet1000 vs. ImageNet100, we were unable to include a verification in the main text due to resource limitations, but we will consider exploring whether similar trends hold in our setup if resources allow.
>
> > Q1. I found Figure 1 confusing. Naively, I would think the L/R flip of the objects would also be preserved as an augmentation, but this stands out as something that the invariance-only model is doing better at preserving than the equivariant model. Could this be explained, or could the figure be made clearer? For instance, what would hypothetical images be that visually contain the augmentation-aware information but do NOT contain semantic structure?
>
> We first note that **Figure 1** illustrates the existence of shared information between representations learned **exclusively** with the invariant or equivariant objective. Both models retrieve samples from the same class, indicating the shared information (e.g., semantic structure), while the *equivariance-only* model additionally preserves augmentation-related information, suggesting that the two objectives rely on a common set of information while capturing complementary aspects.
>
> Nonetheless, we agree that the LR flip shown in retrieval results may be confusing. We conjecture that this implies that the LR flip is considered less important when measuring similarities on learned representations. To confirm that the LR flip is also captured in learned representations, we extended the experiment with LR flipped STL10 test dataset, essentially doubling the number of test data.
>
> We observed that the *equivariance-only* model retrieves samples in a pattern that highlights flip-aware behavior:
>
> - 1-NN: A flipped version of the query image.
> - 2-NN: A sample with a similar pose to the query.
> - 3-NN: A flipped version of the second image.
> - A similar trend is observed in the pair of 4- and 5-NN, and so on.
>
> In addition, the *equivariance-only* model retrieves a greater number of neighbors that share both consistent color and similar pose compared to the *invariance-only* model.
>
> This pattern suggests that the model captures flip-related variation in representation. In contrast, the invariance-only model tends to retrieve samples with random pose, highlighting its preference for transformation-agnostic similarity.
>
> > Q3. How does this learned representation (the backbone representation) relate to the assumption of independence of the projectors? Naively, I am not sure why having completely independent projectors should encourage one to learn a better representation, and as the text is currently written, the more general motivation for benefits of this change are not clear.
>
> We would like to clarify how the independence or separation of the projection heads affects the quality of the learned backbone representation. Our analysis is based on two complementary perspectives:
>
> 1. Redundant gradients from independent heads interfere with task-specific learning.
>
> When the invariant and equivariant heads are completely independent projectors, both must extract shared factors separately. This leads to redundant supervision on shared components, making the backbone repeatedly optimize for the same features. Consequently:
>
> - Model capacity is used inefficiently.
> - Task-specific gradients become weaker, as redundant gradients dominate.
> - The expressivity of the backbone is limited.
>
> We empirically confirmed this by computing the cosine similarity between gradients from different heads with respect to the backbone: it decreased from **0.59** (EquiMod [1]) to **0.34** (Single Shared Projection) and **0.30** (MMoE Projection), indicating reduced redundancy and improved specialization.
>
> 2. Explicit disentanglement leads to easier optimization and faster convergence.
>
> When each head is forced to handle both shared and task-specific information simultaneously, optimization becomes more difficult. By introducing a shared pathway (e.g., a shared projector or MMoE gating), each head can specialize more effectively, leading to better optimization signals. This improves:
>
> - Gradient quality early in training, when learning rate is high.
> - Convergence speed and stability of head parameters.
>
> To support this, we measured the Frobenius norm of weight changes in projectors every 10 epochs. Our method showed smaller changes compared to EquiMod [1], indicating faster convergence of each head.
>
> | Method        | 20   | 30   | 40   | 50   | 60   | 70   | 80   | 90   | 100  | 110  | 120  | 130  | 140  | 150  | 160  | 170  | 180  | 190  | 200  |
> | :-: | :-:  | :-:  | :-:  | :-:  | :-:  | :-:  | :-:  | :-:  | :-:  | :-:  | :-:  | :-:  | :-:  | :-:  | :-:  | :-:  | :-:  | :-:  | :-:  |
> | EquiMod       | 7.16 | 5.05 | 4.19 | 3.74 | 3.47 | 3.27 | 3.07 | 2.88 | 2.66 | 2.43 | 2.18 | 1.91 | 1.63 | 1.33 | 1.02 | 0.72 | 0.44 | 0.19 | 0.04 |
> | Single Shared | 6.58 | 4.61 | 3.86 | 3.54 | 3.27 | 3.13 | 2.98 | 2.81 | 2.58 | 2.37 | 2.10 | 1.85 | 1.59 | 1.30 | 1.00 | 0.71 | 0.42 | 0.19 | 0.04 |
> | MMoE          | 5.28 | 3.13 | 2.21 | 1.88 | 1.68 | 1.55 | 1.45 | 1.35 | 1.25 | 1.15 | 1.03 | 0.91 | 0.78 | 0.64 | 0.50 | 0.36 | 0.22 | 0.10 | 0.02 |
>
> These two analyses collectively demonstrate that our design not only reduces redundant supervision but also facilitates more effective and stable optimization, leading to improved representation.
>
> ### Reference
> [1] Devillers and Lefort. EquiMod: An Equivariance Module to Improve Self-Supervised Learning. In *ICLR*. 2023.
>
> [2] Chen et al. Exploring Simple Siamese Representation Learning. In *CVPR*. 2021.
>
> [3] Yerxa et al. Contrastive-Equivariant Self-Supervised Learning Improves Alignment with Primate Visual Area IT. In *NeurIPS*. 2024.

---

> > ### Comment · Reviewer_TqVZ · 2025-08-04
> >
> > Thank you for the clarifications, analyses, and pointers to specific places in the paper. I particularly appreciated the analysis of the gradients during optimization using two different projectors. This is an interesting way to consider how the independence of projectors can influence the backbone.
> >
> > Overall, would it be possible to summarize the changes that will be made to the paper to address each of these points? Some were mentioned in the rebuttals, but I wasn't sure exactly what would be changed to address W1, W2, Q1 and Q3. I will reevaluate my score once the proposed changes are clear. A few additional comments and suggestions below.
> >
> > Re W1/Q1: Thank you for the clarification here, but it is still not totally clear to me how the results in Figure 4 apply to the discussion on lines 39-46 about invariant-only vs. equivariant-only models. I believe the baseline model shown (EquiMod) is one that has both an equivariant and invariant head. This confusion could likely be addressed by refining the writing and motivation, because this also relates to Figure 1 and the confusion with that figure (thank you for the additional analysis of the K-NN here, I'm assuming it is something about pooling that will cause the L/R flips to be essentially represented as the same thing).
> >
> > Re W2: I understand that the gating weights show the experts are doing different things, but the idea that these are "invariant" and "equivariant" is not quantified in the paper (this is just qualitatively shown with the K-NN embeddings). This was also raised by reviewer inzh, especially because an "equivariant" representation has a specific definition -- it looks like some analysis has been done in response to Reviewer inzh which begins to address this by doing a regression analysis to predict the augmentation parameters, so this should at minimum be included in the paper (and possibly extended, as inzh discusses).

---

> > > ### Author Response · Authors · 2025-08-06
> > >
> > > Thank you for further engaging in the discussion. Below we summarize the tentative changes corresponding to each of your concerns. If you have additional suggestions, we would be happy to take them into account.
> > >
> > > Regarding **W1**: We acknowledge the concern about quantifying representational redundancy. While this was presented in Figure 4, the relationship between redundancy in the projector and its influence on the learned encoder was not clear. We will clarify this connection around Figure 4 (the paragraph **Redundancy Across Experts**), and incorporate the gradient analysis in our response to **Q3**.
> > >
> > > Regarding **W2**: **Figure 3** presents both **quantitative** and **qualitative** evidence of expert specialization. However, the qualitative aspect may not be clear, as the paragraph **Expert Specialization** referring to Figure 3 does not explicitly discuss the quantified results. We will also try to revise the overall paragraph to better balance the discussion of both **quantitative** and **qualitative** aspects. In addition, we will provide a similar analysis on ImageNet100 in Appendix.
> > >
> > > Regarding **Q1**: The LR flip causes confusion, as it is one of the most visually salient transformations for humans, while the learned representations seem to prioritize other aspects. We will expand the retrieval dataset to include LR-flipped images and add a brief discussion on this behavior, tentatively in a footnote.
> > >
> > > Regarding **Q3**: We appreciate Reviewer TqVZ for raising this concern, and we believe the gradient analysis further strengthens our contribution. As stated above for **W1**, this analysis will follow **Figure 4**, as the discussion on representational redundancy in projectors is naturally followed by its effect on encoders.
> > >
> > > Below we address your additional comments and suggestions:
> > >
> > > > Re W1/Q1. Thank you for the clarification here, but it is still not totally clear to me how the results in Figure 4 apply to the discussion on lines 39-46 about invariant-only vs. equivariant-only models. I believe the baseline model shown (EquiMod) is one that has both an equivariant and invariant head. This confusion could likely be addressed by refining the writing and motivation, because this also relates to Figure 1 and the confusion with that figure (thank you for the additional analysis of the K-NN here, I'm assuming it is something about pooling that will cause the L/R flips to be essentially represented as the same thing).
> > >
> > > First, we would like to clarify that **Figure 1** and **Figure 4** are not directly connected and serve different purposes in the paper.
> > >
> > > **Figure 1** is intended solely as a motivational example using ablated models. The *invariance-only* model corresponds to SimCLR, while the *equivariance-only* model refers to *EquiMod without the invariant loss for equivariant learning*, as noted in **line 41**. (We will clarify this in the caption to avoid potential misunderstanding.) The *equivariance-only* model is trained using only the equivariant head and objective, without any invariant head or loss. Despite this, **Figure 1** shows that the model can retrieve semantically consistent samples, indicating that semantic information can also be captured through equivariant learning, which was considered as an ability attributed to invariant learning.
> > >
> > > This observation suggests that invariant and equivariant heads may learn redundant information, especially in the setting with only two independent projectors (e.g., EquiMod). We intended this figure to motivate the possibility of redundancy at the projector-level.
> > >
> > > **Figure 4**, on the other hand, focuses on validating that our proposed MMoE projection can reduce this redundancy. Using canonical correlation, we quantify the redundancy between projectors and show that compared to EquiMod, which uses two independent projectors, our method achieves lower redundancy and improved out-domain performance.
> > >
> > > In the revision, we will refine the main text to better distinguish the intent of **Figure 1** and **Figure 4** (or even not referring to **Figure 4** here).
> > >
> > > (Continue to the next)

---

> > > ### Author Response · Authors · 2025-08-06
> > >
> > > > Re W2. I understand that the gating weights show the experts are doing different things, but the idea that these are "invariant" and "equivariant" is not quantified in the paper (this is just qualitatively shown with the K-NN embeddings). This was also raised by reviewer inzh, especially because an "equivariant" representation has a specific definition -- it looks like some analysis has been done in response to Reviewer inzh which begins to address this by doing a regression analysis to predict the augmentation parameters, so this should at minimum be included in the paper (and possibly extended, as inzh discusses).
> > >
> > > The gating weights directly reflect the role of each head, e.g., larger weights assigned to the equivariant loss during optimization indicate specialization toward capturing equivariance. However, we agree that quantifying invariance/equivariance through other metrics would further substantiate this interpretation.
> > >
> > > To quantify whether each expert specializes in invariance or equivariance, we adopt the parameter equivariance (P-equivariance) evaluation protocol from a concurrent work [1], which learns predictors on top of the frozen backbone and experts for a downstream task of *predicting the transformation parameters applied in the input space by comparing the embeddings of the original and transformed images* to measure whether the transformation parameter is predictable by comparing them on the latent space. (Obviously, the transformation parameter is not predictable from perfectly invariant representations.)
> > >
> > > Concretely, we freeze the backbone and experts, then train predictors that take **the embedding of the original and transformed images** as inputs, and predict **the transformation parameters**. We measure the MSE of the transformation parameter between the predicted and true transformation parameters as the P-equivariance measure, and the cosine similarity between the expert-level embeddings of the original and transformed images is as the invariance measure.
> > >
> > > | Index | P-equivariance ↓ | Invariance ↑ |
> > > | :-: | :-: | :-: |
> > > | 1 (Shared)  | 0.70 | 0.94 |
> > > | 2 (Inv.) | 0.81  | 0.94 |
> > > | 3 (Inv.)  | 0.80 | 0.94 |
> > > | 4 (Inv.) | 0.80 | 0.93 |
> > > | 5 (Inv.) | 0.79 | 0.94 |
> > > | 6 (Inv.) | 0.77  | 0.93 |
> > > | 7 (Equiv.) | 0.37  | 0.57 |
> > > | 8 (Equiv.) | 0.27  | 0.79 |
> > >
> > > The results show clear differences in invariance and equivariance scores across experts, suggesting that their roles are clearly separated. This provides quantitative support for our interpretation of expert specialization, complementing the qualitative evidence in the original submission. We will include this analysis in the revised manuscript to clarify the role of experts more rigorously.
> > >
> > > ### Reference
> > > [1] Plachouras et al. Towards a Unified Representation Evaluation Framework Beyond Downstream Tasks. In *IJCNN*. 2025.

---

> > > > ### Comment · Reviewer_TqVZ · 2025-08-09
> > > >
> > > > Thank you for the summarization of changes and the additional equivariant/invariant evaluation. The presented analysis in the rebuttal strengthens the results, and puts the paper into better context with other work. I also think that the suggested changes to clarify the paper and the figures would greatly improve the paper. I have updated my score accordingly.

---

> > > > > ### Author Response · Authors · 2025-08-09
> > > > >
> > > > > Thank you for your helpful feedback, especially regarding Figure 1 and the invariance/equivariance evaluation. We will revise our manuscript accordingly and include additional analyses to further strengthen our contribution.

---

### Note · Authors · 2025-08-14

First of all, we appreciate your time and effort in reviewing our paper.

Our work introduces a task-aware routing mechanism based on MMoE that dynamically allocates shared and task-specific experts to reduce redundancy between invariant and equivariant learning. This design makes more effective use of model capacity, resulting in better downstream performance and generalization through reduced redundancy and clear expert specialization

Reviewers noted the method’s ability to dynamically allocate shared and task-specific experts with reduced redundancy across projectors (TqVZ S1, inzh S4), its broad compatibility with various SSL frameworks (TqVZ S2, inzh S2, tSPH S3), and its extensive validation over diverse datasets and tasks (inzh S3, 5sNH S2, tsPH S1), along with strong analyses on redundancy reduction and expert specialization (inzh S4, 5sNH S3&S4).

Below we summarize major changes we will incorporate in the revision, in response to the reviewers’ suggestions:

**1. Relationship between redundancy in projectors and learned representations (TqVZ Q3, 5sNH W4)**

We analyzed why redundancy across the projectors during training can negatively affect the backbone, focusing on gradient and weight differences among projectors.

**2. Matching computational cost conditions (5sNH W2)**

We conducted additional experiments under matched computational cost conditions to verify that the benefits extend beyond out-domain performance to other tasks.

**3. Quantitative evaluation of equivariance (TqVZ W2, inzh W2)**

We provided quantitative evaluations of equivariance using R- and P-equivariance metrics in both learned representations and the outputs of individual experts.

**4. Asymmetric expert sharing (inzh W1)**

We clarified asymmetric specialization naturally emerges as the gating mechanism allocates experts to invariant and equivariant objectives while avoiding redundant learning across projectors, and showed that the improved performance is related to reduced redundancy achieved by our method.

**5. Introduction and Figure 1 (TqVZ Q1, tSPH W1–3)**

We will revise the manuscript to strengthen the presentation of the motivation, and update Figure 1 to avoid confusion from the LR flip, e.g., by including the flipped images in the retrieval set.

**6. Analysis of expert specialization on ImageNet100 (TqVZ W2)**

We will extend our analysis of expert specialization to the ImageNet100 pretrained model.

Thank you very much.

Authors

---

### Decision · Program_Chairs · 2025-09-17

**Decision:**

Accept (poster)

**Comment:**

This paper studies the problem of representational self-supervised learning based on invariant and equivariant objective functions, for which the work introduces a task-aware routing method that is based on the Mixture of Experts framework. The reviewers have all expressed support for the paper, with both Reviewer TqVZ and inzh noting the timeliness and importance of the proposed MMoE approach. I therefore recommend acceptance, though I strongly encourage the authors to incorporate the suggestions that arose during the discussion period, since as pointed out by the reviewers, these greatly strengthen and clarify the results of the work.